# Quantification of uncertainty in 3-D seismic interpretation: implications for deterministic and stochastic geomodelling and machine learning

Alexander Schaaf[1] and Clare E. Bond[1]

[1]Geology and Petroleum Geology, School of Geosciences, University of Aberdeen, AB24 3UE, UK

**Correspondence:** a.schaaf@abdn.ac.uk

**Abstract.** In recent years uncertainty has been widely recognized in geosciences, leading to an increased need for its quantification. Predicting the subsurface is an especially uncertain effort, as our information either comes from spatially highly limited direct (1-D boreholes) or indirect 2-D and 3-D sources (e.g. seismic). And while uncertainty in seismic interpretation has been explored in 2-D, we currently lack both qualitatitive and quantitative understanding of how interpretational uncertainties of 3-D datasets are distributed. In this work we analyze 78 seismic interpretations done by final year undergraduate (BSc) students of a 3-D seismic dataset from the Gullfaks field located in the northern North Sea. The students used Petrel to interpret multiple (interlinked) faults and to pick the Base Cretaceous Unconformity and Top Ness horizon (part of the Mid-Jurassic Brent Group). We have developed open-source Python tools to explore and visualize the spatial uncertainty of the students fault stick interpretations, the subsequent variation in fault plane orientation and the uncertainty in fault network topology. The Top Ness horizon picks were used to analyze fault offset variations across the dataset and interpretations, with implications for fault throw. We investigate how this interpretational uncertainty interlinks with seismic data quality and the possible use of seismic data quality attributes as a proxy for interpretational uncertainty. Our work provides a first quantification of fault and horizon uncertainties in 3-D seismic interpretation, providing valuable insights into the influence of seismic image quality on 3-D interpretation, with implications for deterministic and stochastic geomodelling and machine learning.

## 1 Introduction

Geosciences, and geology in particular, are concerned with integrating various sources of data, often of limited, sparse and indirect nature, into scientific models. The use of limited data combined with our limited knowledge of the highly complex earth system, invariable infuses any model with uncertainty. Especially as geology inherently relies heavily on interpreted data that often requires reasoning about processes that occur over geological time scales (Frodeman, 1995), which further increase the space of uncertainty.

The interpretation of 3-D seismic data is an integral part of constructing structural geomodels of the subsurface and plays a major role in the energy industry. Due to the indirect, noisy and non-unique nature of seismic data processing into images, interpretation is inherently uncertain. Our work is thus concerned with quantifying the scope of uncertainties in seismic interpretation, which represents inevitably biased, human judgment under uncertainty (Tversky and Kahneman, 1974). This "subjective" uncertainty is in contrast to more "objective" uncertainty related to the geophysical acquisition of the data itself (Tannert et al., 2007; Bond, 2015). Previous work has shown that significant conceptual uncertainties and biases are encountered during the interpretation process of 2-D seismic lines (Bond et al., 2007, 2011; Macrae, 2013; Bond, 2015; Alcalde et al., 2017a, c), as well as the impact of seismic image quality on the interpretation (Alcalde et al., 2017b). But subsurface structures are naturally three-dimensional and the use of 3-D seismic data is ubiquitous in industry (Biondi, 2006). This raises the need to further our understanding of the distribution of interpretational uncertainties in 3-D space (Abrahamsen et al., 1992; Thore et al., 2002; Thiele et al., 2016; Godefroy et al., 2018). Additionally, the process of interpretation between seismic lines and cubes is fundamentally different, and thus might lead to conceptually different uncertainties to be dominant (e.g. the need to connect fault evidence between seismic lines introduces significant uncertainty in widely spaced 2-D interpretation compared to fault interpretation in seismic cubes; see Freeman et al., 1990).

In this work we investigate the scope of uncertainties in 3-D seismic interpretation. We qualitatively and quantitatively analyze interpretations of 78 final year undergraduate students (BSc) conducted on a 3-D seismic cube of the Gullfaks field. The dataset depicts a comparatively simple geometry of planar domino-style normal faults, but the seismic dataset exhibits high amounts of noise, especially in its eastern half, and generally increasing with depth and in fault proximity (see Fig. 8A, B). This inhibits straight-forward interpretation of major faults and horizons and limits use of structural seismic attributes. We analyze the spatial variation in fault stick interpretations and the subsequent uncertainty in fault orientation. Horizon interpretations are analyzed and combined with fault interpretations for a description of fault throw uncertainty. Additionally, we investigate the differences in fault network topology (see Morley and Nixon, 2016; Peacock et al., 2016, 2017) in the interpretation ensemble to better estimate the uncertainty of interpreting fault networks in 3-D seismic data. We use the interpretation of Fossen and Hesthammer (1998) as a reference expert example (in the sense of Macrae et al., 2016) to compare fault network topology and fault orientation uncertainty with the student interpretations. We integrate our findings of interpretation uncertainties with its relation to seismic data quality and discuss the implications for both deterministic and stochastic geomodelling, as well as machine learning applied to seismic interpretation.

## 2  Materials and Methods

### 2.1  Gullfaks geology and seismic data

We give here a brief overview over the regional and structural geology of the study area; a more in-depth description of the structural geology of the Gullfaks Field can be found in Fossen and Hesthammer (1998).

The Gullfaks Field is a subset of the NNE-SSW-trending, $10 - 25$ km wide Gullfaks fault block, located in the western part of the Viking Graben within th northern North Sea (see Figure 1a). The Gullfaks field's reservoir units reach from the

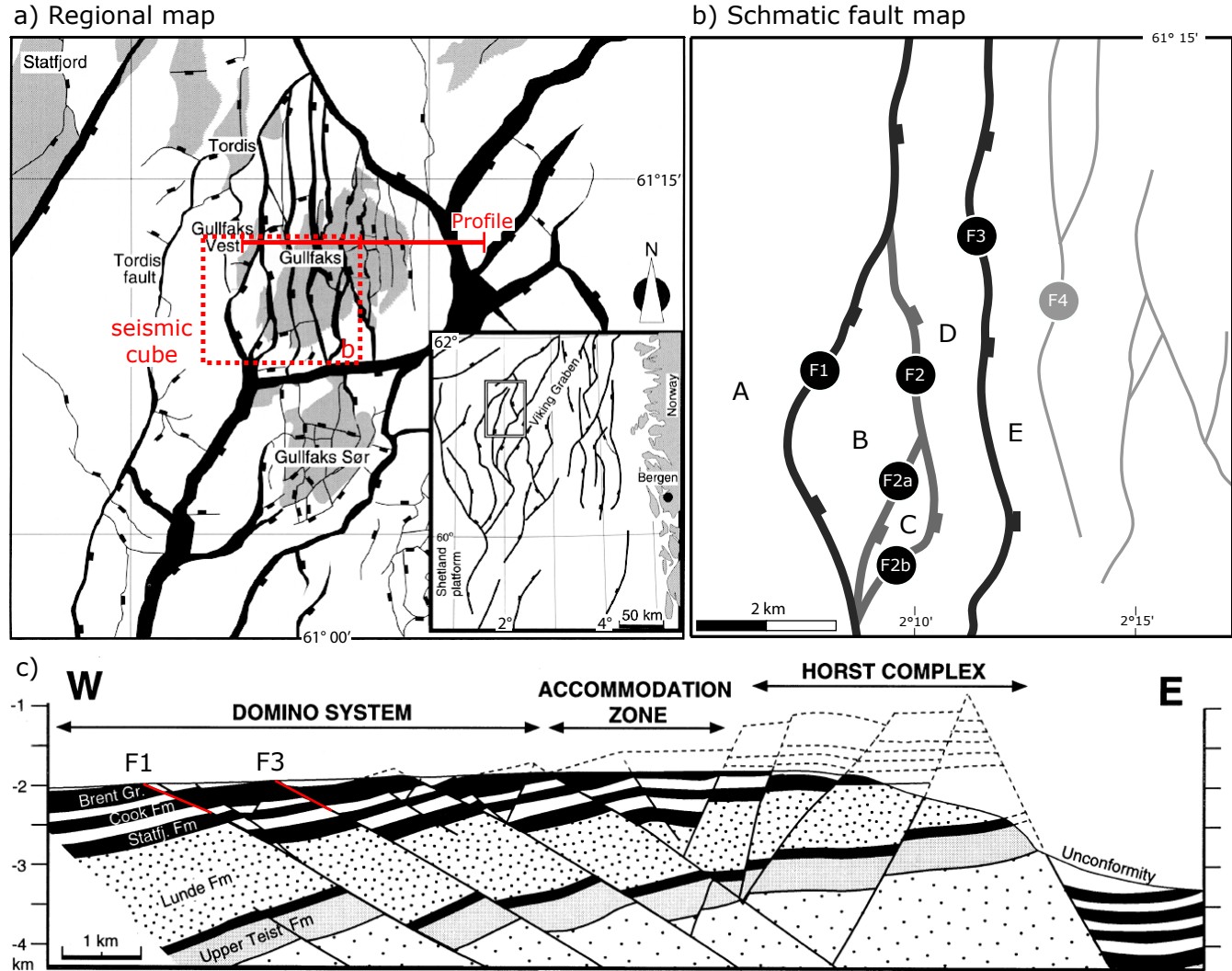

**Figure 1.** (a) Regional overview of the Gullfaks-Statfjord area located in the northern North Sea, showing the location of the Gullfaks oil field; (b) Fault map of the Statfjord Formation, showing the main faults of the Gullfaks field and their labeling; (c) Cross-section across the Gullfaks field, depicting the three distinct structural domains, major faults and stratigraphy (modified from Fossen and Hesthammer, 1998).

late Triassic Hegre Group, over the Early Jurassic Statfjord Formation, Dunlin Group up to the Brent Group (Hesthammer and Fossen, 1997). The reservoir units are separated from the Upper Cretaceous sediments above by the Base Cretaceous Unconformity (Fossen and Hesthammer, 1998). The field consists of three structurally distinct domains: a structurally simple domino system in the western part, and the structurally more complex accommodation zone and Horst complex towards the east (see Figure 1c). Our study focuses on the structurally simpler western part of the domino system, where we investigate the

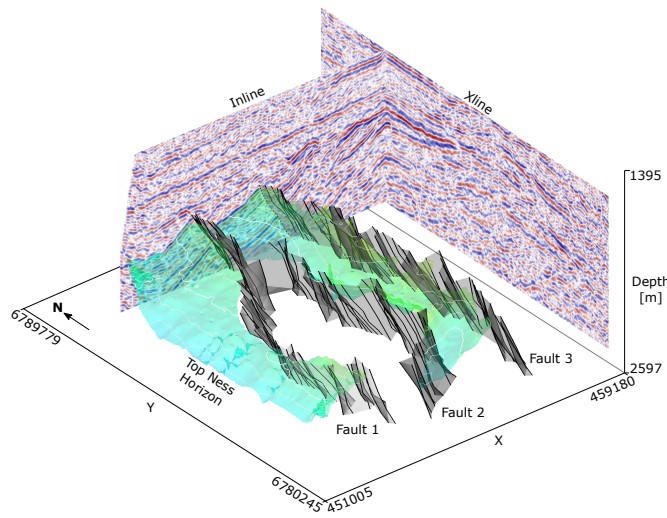

**Figure 2.** Example interpretation from a single student, showing the three major faults considered in our study as well as part of the Top Ness horizon interpretation. BCU and additional faults towards the East are hidden for better visualization of the key elements.

uncertainty of the three faults F1 - F3 and the fault blocks A - E depicted in Figure 1b. Note that Fault 1 and 2 merge in the northern half and at the bottom of the domain and that Fault 2 splits into two smaller faults F2a and F2b.

The 3-D seismic survey of the Gullfaks field, ST85R9211, was recorded in 1985 and reprocessed in 1992. It was recorded in time and converted to depth using TWT depth conversion and was migrated using a Prestack Kirchhoff migration.

## 2.2 Interpretation dataset

The analyzed interpretations were produced as part of the Surface and Subsurface Digital Imaging course within the undergraduate program Geology and Petroleum Geology at the University of Aberdeen. While the students had prior training in structural geology and interpretation of 2-D seismic data, this was the student's first hands-on course in 3-D seismic interpretation using the Petrel software as part of their undergraduate program. The fourth year undergraduate (BSc) students loaded the seismic data into Petrel together with 14 wells, while ensuring proper georeferencing. The following interpretation process focused on first interpreting the Top Cretaceous horizon with initial support by the lecturing staff. Afterwards the students started to independently interpret the Base Cretaceous Unconformity (BCU) and Top Ness horizon (which is part of the Brent group) around well locations, followed by connecting the horizon interpretation in-between wells. The students were instructed to mainly use guided auto-tracking, as well as occasional seeded tracking and manual interpretation where possible or necessary, depending on seismic data quality. Afterwards fault interpretations were conducted of major faults. The students then interpolated surfaces from the horizon interpretations using Petrel's *Make Surface* function. Polygons were created based on fault locations to create a Top Ness surface subdivided into the fault blocks.

For our study we collated the interpretation data from the students Petrel projects into a joint project, where interpretations were sorted and labeled. Of a total of 90 student interpretations, we used 78 in our study, as the other 12 either lacked relevant interpretations or were corrupted Petrel project files. Interpreted Top Ness horizon surfaces and fault sticks where then exported systematically to allow for automated data processing and analysis. An example student interpretation is shown in Fig. 2, containing the three major faults considered, as well as the Top Ness horizon in two of the fault blocks A and D defined in Fig. 1b.

## 2.3 Data analysis

To process and analyze the large amount of interpretation data, the exported Petrel surfaces and fault sticks were wrangled using custom Python functionality and labeled in an open tabular data format. The result is a set of 4460878 data points, belonging to 78 student interpretations, with 228 unique faults considered, which consist of a total 10052 individual fault sticks. For data processing and analysis we made heavy use of the open-source Python packages *pandas*, *SciPy* and *NumPy* (McKinney, 2011; Jones et al., 2001; Oliphant, 2006).

For the purpose of visualization and statistical analysis of the fault interpretations across the collective interpretations, the domain was discretized into regular bins ($nx = 60$, $ny = 60$, $nz = 24$). In the following analysis we present 2-D and slices of 3-D histograms of fault interpretations, showing interpretation frequency across the domain. Fault orientations for individual faults were computed by fitting a plane (using singular-value decomposition) to all fault stick points of a single interpretation over all grid cells (collapsed along the x-axis). This makes the analysis less dependent on the fault stick interpretation density, which varies extensively in-between students. The resulting normal vector can be converted into strike and dip values. For visualization the fault orientation data was subdivided into three regular bins oriented E-W across the structures. The fault throw analysis is based on the Top Ness horizon and is computed individually at each interpreted fault stick for each fault and each interpretation. The fault throw for each interpretation is then averaged into 20 bins along the Y-axis to make the analysis independent of the number of fault sticks interpreted by each student. The nearest data points on both the hangingwall and footwall of the Ness horizon fault blocks were selected as seeds. From these seeds the surface data approximately orthogonal to strike was used within a strike-parallel window of three grid cells (approx. $570~m$). A relative gradient filter was then used to exclude points with gradients to their nearest neighbors outside of the inner-quartile-range (IQR) of the selected subset. The resulting data is fitted with a linear regression for each fault block and the intersection with the fault stick used to calculate fault throw.

Throughout this work we make use of the term seismic data quality not in the strictly geophysical sense of quality factors surrounding seismic data acquisition and processing, but rather in the sense of the interpretability of the seismic data. If the seismic data lacks clear, continuous reflectors in a region, but shows a noisy image difficult to interpret—no matter what the source of this may be—we describe it as an area of low seismic data quality. Similarly, if reflector strengths are high and continuous (for horizon interpretation) or clearly offset (for fault interpretation), we speak of high seismic data quality.

For the assessment of seismic reflector strength an RMS Amplitude seismic attribute (RMSA for short) was calculated using Petrel's RMS Amplitude function using a window length of 24 traces. RMS Amplitude represents the square root of the arithmetic mean of squared amplitude values across a specified seismic trace window.

# 3 Results

## 3.1 Uncertainty in fault interpretation

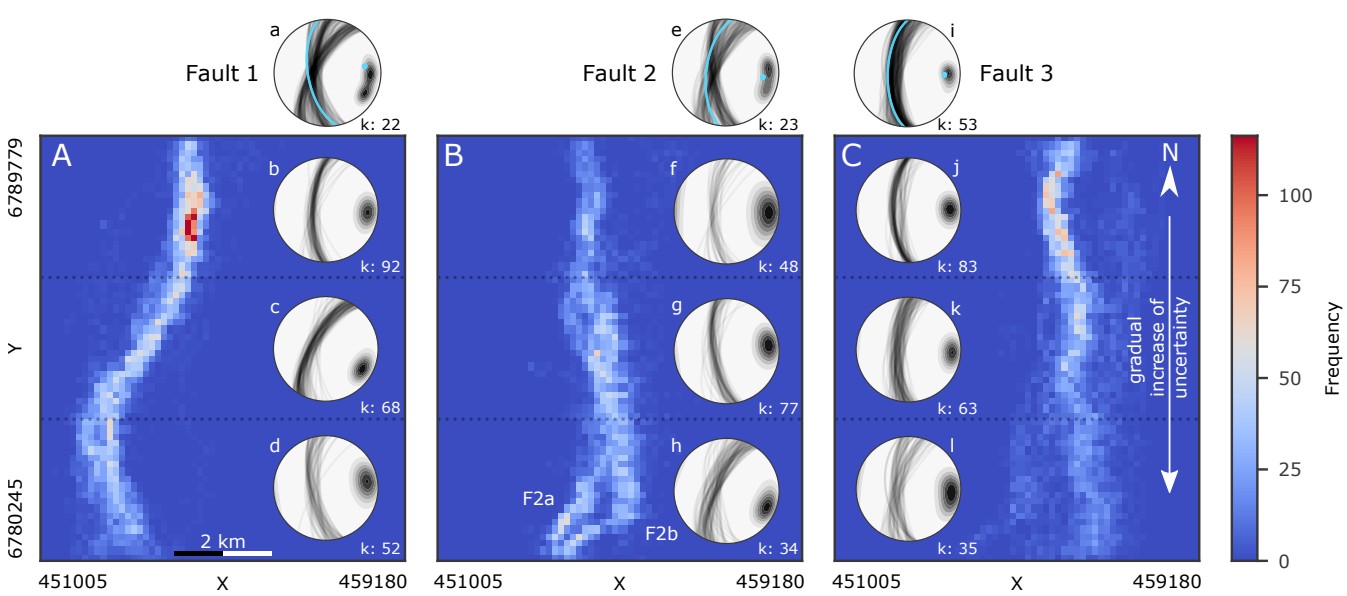

**Figure 3.** 2D Histograms for the three Faults 1 (A), 2 (B) and 3 (C) for depth slice at $2\ km\ \pm\ 0.1$. Stereonet plots of Faults 1-3 (columns), with all fault strike orientations along the fault length plotted combined in the first row (a, e, i). To discriminate changing trends in actual fault orientation from interpretation uncertainty rows two to four plot data from bins separated by dashed blue lines, from the northern bin (c, g, k), middle (d, g, l) and south (e, h, m). Blue planes and poles in top row (a, e, i) show Bingham analysis mean pole from Fossen and Hesthammer (1998) for reference.

Figure 3 shows 2-D histograms for the three major faults taken into consideration within this study. The histograms cover the entire extent of the seismic cube, with the frequency of fault stick points counted per bin in a depth slice at $2\ km\ \pm\ 0.1$. Fault 1 shows a sigmoidal shape in the N-S direction of the seismic depth slice (see Fig. 3A), with high frequency densities in the northern part and lower intensities found in the southern part. Plotting all fault plane orientations within a single stereonet reveals three distinct clusters of planes (Fig. 3a), which when separated into three equal bins along the N-S axis correspond to the components of the sigmoidal fault shape (Fig. 3b-d). We have added Bingham mean poles from Fossen and Hesthammer (1998) for all three faults in the plot (light blue) for comparison. Interpretations of Fault 2 show a split into two sub-faults F2a and F2b in the southern part (see Fig. 3B), as also interpreted by Fossen and Hesthammer (1998, see Fig. 1). The spatial

uncertainty of the fault interpretations appears slightly lower in the northern part of the seismic slice, but also shows evidence of a small separated fault block in the central part of the fault. The interpretations of Fault 3 show a strong increase in dispersion towards the southern part of the seismic slice (Fig. 3C). The same trend of increasing uncertainty can be observed in the fault plane orientation (Fig. 3j-l). Additionally, the histogram shows the occasional interpretation of the fault branching towards Fault

5  2(b), towards the West, and towards Fault 4 to the East (see Fig. 1b). The effect of fault stick interpretation frequency between the students with below-median and above-median fault stick interpretation frequency on overall fault standard deviation was analysed using Bayesian estimation for two groups (Kruschke, 2013). We observed a difference in mean standard deviation of $35.8\ m$, $20.2\ m$ and $81.5\ m$ for Fault 1, 2 and 3 respectively, with probability of the differences being larger than zero being $99.3\ \%$, $87.4\ \%$ and $99.9\ \%$ respectively, making the differences for Fault 1 and 3 statistically credible.

10     Figure 3 plots fault plane orientations calculated from students fault stick interpretations. Fault 1 shows three distinct clusters of orientations (Fig. 3A, a-d), which can be separated by subdividing the study domain into three equal bins along the N-S axis. Fault 1 shows a striking decrease in k-values (a measure of tightness of the orientation clusters) from North to South (92, 68, 52). The pattern does not hold true for Fault 2, which shows low k-values both in the North and South. Fault 3 shows similar behaviour to Fault 1, with a strong increase in dispersion from North to South (83, 63, 35).

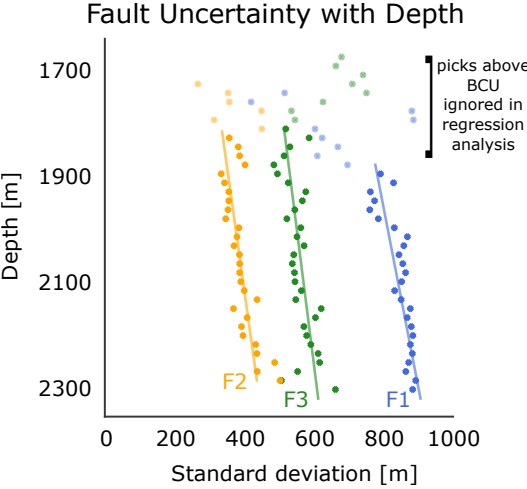

**Figure 4.** Scatter plot of mean (collapsed y-axis) standard deviation along x-axis of mean fault surfaces for Fault 1, 2 and 3. Chaotic patterns of uncertainty at shallow depths (faded data points) are likely due to sporadic numbers of interpretations, as students often stopped interpreting the faults before reaching the Base Cretaceous Unconformity (BCU). Overall, uncertainty of fault interpretations is increasing linearly with depth (linear regression only takes into account interpretations below the BCU, R-values for Fault 1, 2 and 3 respectively: 0.75, 0.85 and 0.61).

15     Overall the observed uncertainty (standard deviation) of the fault plane along the W-E axis appears to be increasing linearly with depth for all three faults (Fig. 4). Note that the overall mean standard deviations between the faults vary greatly: 791, 384 and $575\ m$ for Faults 1, 2 and 3, respectively. The extremely high variation of standard deviations seen in the upper part of Fig.

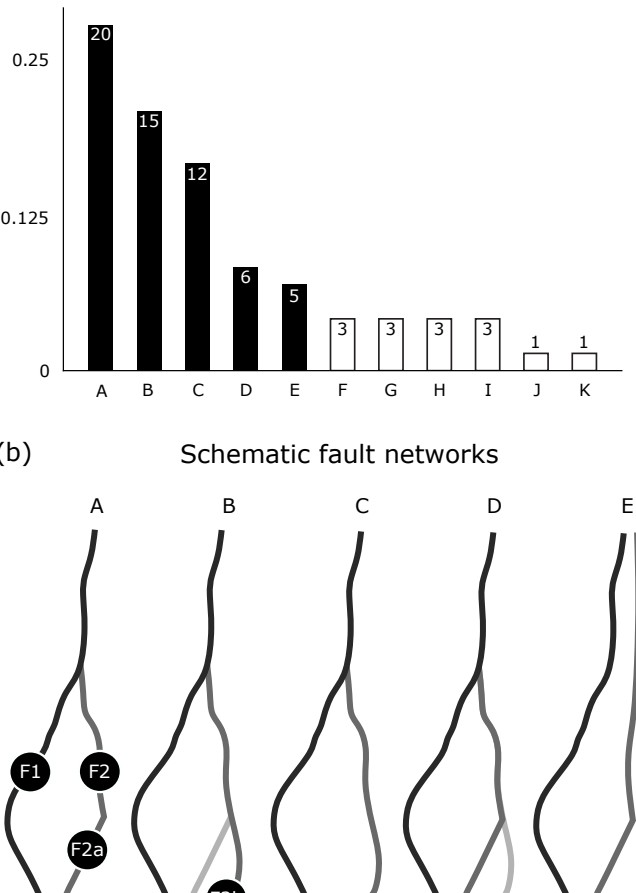

**Figure 5.** Probabilities of unique fault network topologies (a) with corresponding schematic fault networks (b) of the five most likely networks.

4 (faded data points) is due to a few students extending their fault interpretations above the Base Cretaceous Unconformity (BCU), making the data points at that depth statistically unreliable due to low sample numbers and geologically questionable. Any fault stick interpretations above the BCU were thus excluded from the least-squares linear regression (R-values for Fault 1, 2 and 3: 0.75, 0.85 and 0.61).

5     The ensemble of interpretations show 11 different fault network (FN) topologies (see Fig. 5a). Five modes of FN topology make up the bulk of fault network topologies, while others were only interpreted by 3 or less students respectively. The sketches in Fig. 5b represent these five most interpreted FNs (Fault 3 is omitted for brevity, as it was interpreted by all students in a similar fashion, and only a single student connected Fault 2 with Fault 3). Note that the most frequent FN (Fig. 5b, A) is different from the reference expert FN interpretation of Fossen and Hesthammer (1998), which corresponds to either the

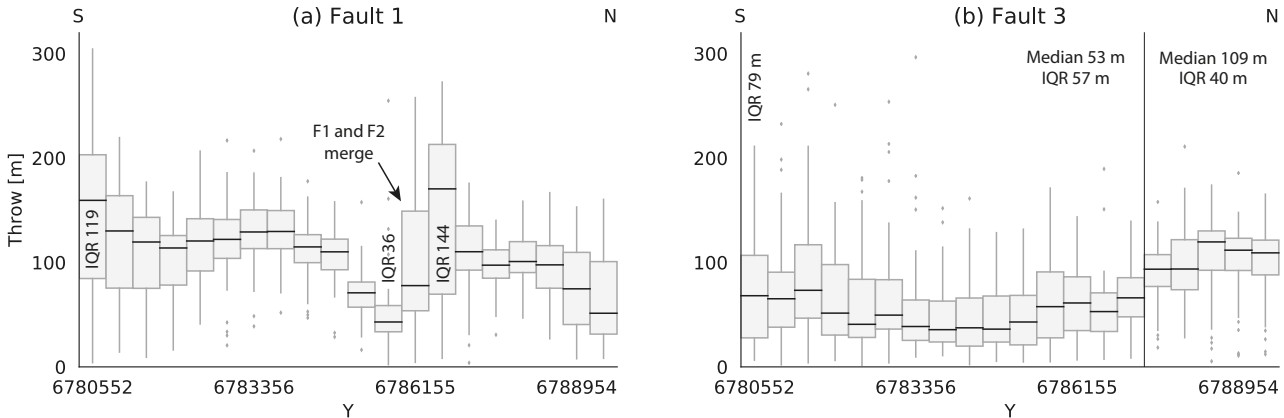

**Figure 6.** Results of fault throw analysis for Faults 1 (a) and 3 (b) visualized as boxplots, showing median fault throw (black lines) with interquartile range (grey boxes), minima and maxima whiskers (grey lines) and outliers (grey dots).

second or fourth most common FN interpretations (Fig. 5b, B and D). The major source of uncertainty in this specific FN appears to be interpreting *both* F2a and F2b, and which one abuts the other. The few students who branched off the southern part of Fault 3 towards the West interpreted Fault 2b as part of Fault 3, but did not connect it to the FN of Fault 1 and 2.

### 3.2 Fault throw and horizon uncertainties

Results of the fault throw analysis are plotted in Fig. 6. The boxplots show median fault throw with the associated interquartile range (IQR), extrema and outlier values along fault strike direction. The throw profile of Fault 1 (Fig. 6a) shows a distinct sinuous shape spatially associated with its interaction with Fault 2. This shows one bin with high median fault throw of approx. $180\ m$ and high fault throw uncertainty (IQR $144\ m$) before strongly decreasing in fault throw values down to a median of about $40\ m$ and one of the lowest IQR along the fault ($36\ m$). Median fault throw then rises steadily towards the South while

also increasing in uncertainty (culmating in an IQR of $119\ m$ in the South). Notice the increase of uncertainty at both ends of the dataset, with increasing median throw in the South and decreasing in the North. The throw profile of Fault 3 (Fig. 6b) shows two distinct levels of throw: In the Northern part of the fault median throw is high at $109\ m$, compared to $53\ m$ in Southern part. IQR increases from $40\ m$ in the North to $57\ m$ in the South, with highest IQR observed at the Southern end of the survey ($79\ m$).

Figure 7a shows the average Top Ness horizon basemap for all interpretations combined. Overall the horizon interpretations are increasing in depth from SE towards the NW of the domain. Figure 7b shows the associated standard deviation of the average Ness horizon interpretation, with an overlay of mean fault intersections and well locations. We observe large horizon uncertainties in vicinity to both Faults 1 and 2 throughout the dataset. An increase of horizon uncertainty is occurring at the southern end of the domain where Faults 1 and 2 are beginning to merge again. In the North the horizon uncertainty

surrounding Fault 3 decreases rapidly with distance from the fault, with two welltops and packages of high reflector strength

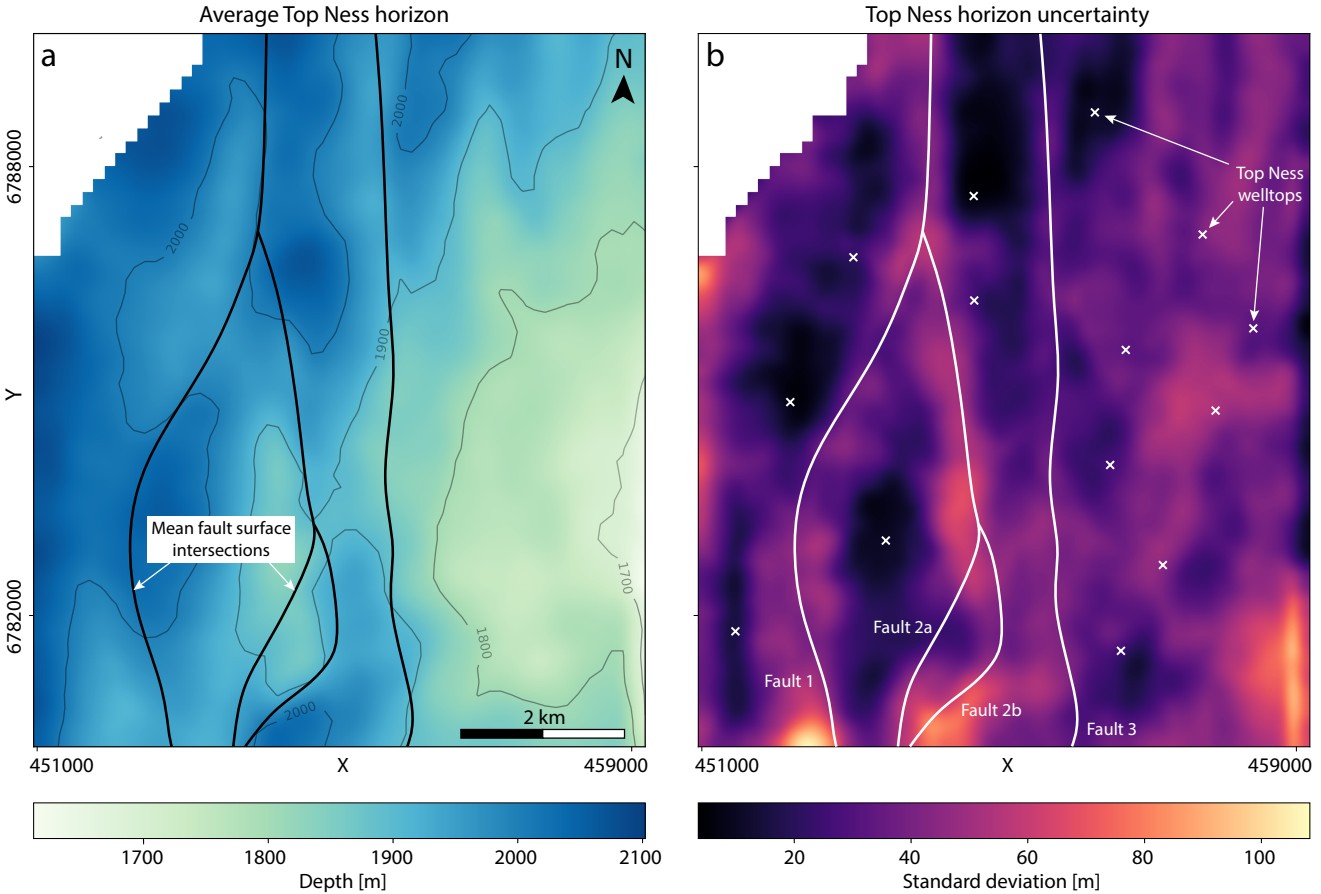

**Figure 7.** (a) Basemap of average Top Ness horizon from all interpretations; (b) Standard deviation of the average Top Ness horizon from all interpretations with overlaid mean fault surfaces (white lines) and welltop locations (white crosses).

(see Fig. 8A) constraining the uncertainty. As the seismic data quality decreases towards the south (see Fig. 8B), the uncertainty in the horizon interpretation increases in the Eastern part of the dataset. Interpretation uncertainties are significantly reduced surrounding well locations in the western part of the study domain, where seismic reflectors of the Top Ness horizon are overall stronger and more continuous (see Fig. 8A). This pattern does not hold true in the East of the dataset, where reflector continuity is overall low and noise in the seismic dataset is high.

### 3.3 Seismic data quality

To assess influences of seismic data quality on fault interpretation uncertainty we made use of the RMS Amplitude (RMSA) attribute as a proxy for reflector strength (strong horizon reflectors aiding the interpretation of faults result in high RMSA values). Specifically, we investigated the example of Fault 3, as it shows significant gradual changes in interpretation uncertainty across the seismic dataset (Fig. 3C and 6b). Figure 8 shows four averaged RMSA responses with corresponding fault stick

interpretation histograms (Fig. 8a-d) from the locations shown as white boxes in Fig. 8.1. In the northern extent of the seismic slice, Fault 3 is closely bounded by strong horizon reflectors, as shown in in the RMSA slice (Fig. 8.1), seismic slice (Fig. 8.2) and inline section A, focusing the students interpretations, as seen in the corresponding histogram of fault interpretations (Fig. 8a). The histogram shows a bimodal distribution, as some students interpreted the fault further towards the East, where another fault is present (Fault 4, see Fig. 1b). The overall uncertainty related to the interpretation of the actual Fault 3 is approximately normal distributed (skewness of 0.33), with the width of the trough seen in the RMSA response containing more than $\pm 1.8$ standard deviations (92.8 %) of the fault stick placement uncertainty. Further towards the south of the dataset, the RMSA response diminishes east of Fault 3, while remaining strong on the western side (Fig. 8b). The fault interpretations show a 64 % increase in standard deviation, with thicker tails in the histogram (79 % increase of Pearson kurtosis), especially towards the West, where interpretations are then bounded by strong seismic reflectors. Further south the seismic response degrades and is noisy (Fig. 8.1 and B), leading also to a lack of signal in the RMSA values (Fig. 8c). The corresponding fault stick placements increase in uncertainty (increase in standard deviation by 289 %), now appearing nearly uniformly distributed with a slight crest (Fig. 8c). At the southern end of the slice, RMSA responses increase again (Fig. 8d). The distribution of fault interpretations shows a bimodal distribution, as seen before in the map view 2-D histograms shown in Fig. 3.

## 4 Discussion

### 4.1 Key findings

Our work has shown that uncertainties in fault stick placement correlate with seismic reflector strengths. In areas of high data constraint this uncertainty is strongly constrained between areas of high RMSA response (Fig. 8a). Our analysis shows how interpretation uncertainty increases with a decrease in data constraints (Fig. 8b). This trend culminates in near-maximal uncertainty in areas of low seismic image data quality (Fig. 8c). The spread in fault stick placement appears to not be entirely driven by seismic noise, but rather appears to be, at least partly, guided by the surrounding interpretations in areas of higher data quality (Fig. 8b, d), allowing interpreted faults to conform to common fault shape models (e.g. to avoid sudden shifts in fault plane orientation). In areas of low data quality the corresponding uncertainty in fault placement is also influenced by more conceptual uncertainties of fault network topology—e.g. interpretations of Fault 3 branching off towards the East or West (as seen in Fig. 3C). Recent research by Alcalde et al. (2017a) has shown the importance fault model availability plays during seismic interpretation (availability bias; see Tversky and Kahneman, 1973, 1974), which should only increase in relevance in areas of low seismic image data quality.

While the domino structure of the study area makes overall tectonic conceptual uncertainty less significant (Bond et al., 2007), the low seismic data quality makes it a challenging interpretation project, as reflector continuity is low and the dataset noisy in large parts of the survey. So despite the increased information density provided by 3-D seismic surveys, significant fault network topology uncertainties remain (Fig. 5).

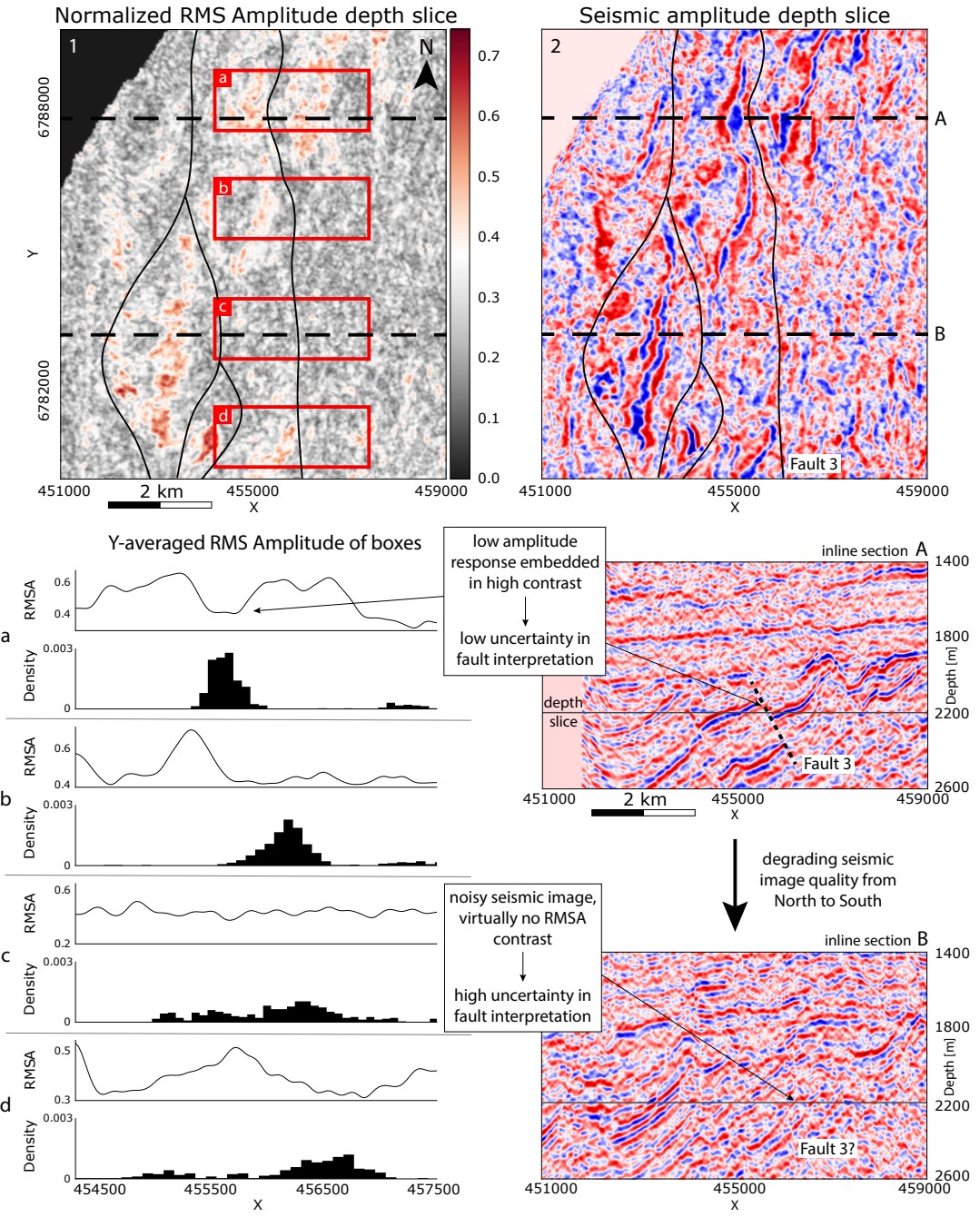

**Figure 8.** Depth slice of RMSA attribute (1) and seismic cube (2) with locations (black horizontal lines) of two inline sections (A, B). Four averaged RMSA responses with corresponding student Fault 3 stick interpretations are plotted in a-d, with their locations noted in the RMSA depth slice (1). The inline sections A and B correspond to the RMSA responses a and b, respectively.

Our study also shows that uncertainties in the placement of faults sticks appears to increase linearly with depth (Fig. 4). This is an important finding for approximating uncertainty trends with depth, and is especially important as seismic image quality tends to decrease with depth.

Our analysis of Top Ness horizon interpretation uncertainties shows the correlation of uncertainty with fault proximity (Fig. 7b). Horizon interpretations surrounding Fault 3 in the Northern part show only slight increases in uncertainty towards the fault, as they are strongly constrained by wells and high reflector continuity (see Fig. 7b and 8A). With decreasing seismic data quality towards the South we also see an increase in uncertainty surrounding Fault 3, which also shows in the increasing fault throw uncertainty seen if Fig. 6b. This trend is not as evident for the throw across Fault 1, with the Top Ness horizon being better constrained on both sides, with overall higher seismic image data quality. Qualitative comparison of median fault throw for both Fault 1 and 2 with depth maps in Fossen and Hesthammer (1998) displaying fault heave can be made under the simplifying assumption of constant fault dip. Fault heave qualitatively mimics the patterns we found in our uncertainty study: gradual decrease from North to South for Fault 3, and the stark differences where Fault 1 and 2 merge, as well as the increase towards the South. This comparison consolidates the confidence in our automated fault throw analysis and hints at the validity of aggregating large number of interpretations of even non-experts to assess geological features.

We have also shown uncertainty in fault orientations (Fig. 3), and the inadequacy of summarizing fault orientation using a deterministic mean pole, as in Fossen and Hesthammer (1998), who calculated it using Bingham analysis of data from a different 3-D seismic dataset. This inadequacy results from sinusoidal fault map pattern (Fig. 3A) and curved 3-D geometries on-top of uncertain fault stick placements.

We observed strong decreases in uncertainty surrounding wells in the West of the study area (Fig. 7b). The trend of increasing uncertainty in horizon location from West to East could be attributed to the decrease in seismic image data quality and the thus much lower reflector continuity. But we would also expect for the horizon uncertainties to be reduced within the immediate surroundings of the wells. One possible explanation for this could be that students focused their interpretation efforts on the higher quality western part of the seismic cube due to the time constraints on the interpretation project.

## 4.2 Implications for deterministic modelling

Our results and analysis suggest that the uncertainties recorded in 2-D seismic interpretation experiments (e.g. Bond et al., 2007; Bond, 2015) are similarly seen in the interpretation of 3-D seismic image data. Akin to the 2-D experiment and analysis of Alcalde et al. (2017b), we show a correlation between seismic image quality and interpretation uncertainty. We have quantified the impact of fault uncertainty on fault network topology and fault and horizon uncertainty on fault throw. The fault network topology in a 'traditional' deterministic geomodel is important as it determines the number of fault blocks and hence the degree to which stratigraphic units are separated (by faults). This information is imperative to the understanding of reservoir compartmentalization in hydrocarbon reservoirs, connectivity and flow characteristics of ground water aquifers and for geothermal projects. Simply, reservoir performance can be significantly affected by fault network topology and understanding these uncertainties, and hence reservoir connectivity, can be critical to the planning of production strategies (e.g. Manzocchi et al., 2008a, b; Lescoffit and Townsend, 2005; Tveranger et al., 2008). The type of fault network topology information avail-

able in Fig. 5 could be used to inform reservoir modelling to guide multiple production strategies (e.g. multiple deterministic models are made) and for informed history matching during field operation when reservoir models have been developed from single deterministic models and need to be updated.

Although not developed in detail in this paper, our analysis of fault throw uncertainties highlight that the use of fault throw information, e.g. to predict fault sealing properties such as shale gouge ratio (Yielding, 2002; Vrolijk et al., 2016), could be significantly affected by uncertainty in the interpretation of 3-D seismic cubes. Our work highlights areas where uncertainties in fault throw are likely to increase: with increasing distance from wells in ares of low reflector continuity, where seismic image quality is poor (and correspondingly with increasing depth), and where faults join or abut. Modelling of uncertainties in fault throw and using information such as that derived here to inform where uncertainties are likely to be greater could provide the basis for more informed modelling of fault seal parameters, such as through stochastic modelling discussed below, or integrated as uncertainty parameters into deterministic geo- and reservoir models. In summary, fault throw uncertainty together with fault network topology uncertainty has the potential to significantly alter predicted fluid-flow patterns in the crust with implications for water-resources, reactive element transfer (e.g. to inform nuclear waste disposal engineering), or hydrocarbon and energy production.

## 4.3 Implications for stochastic modelling

Although we can outline how uncertainty information could be used to better inform use of deterministic models and their inherent uncertainties, advances in both computational capabilities and implicit structural geomodelling have allowed for major improvements in the incorporation of uncertainties into structural geomodels by means of stochastic simulations. At its core stochastic structural geomodelling requires adequate disturbance distributions to obtain reasonable estimates of geomodel uncertainty (see Wellmann and Caumon, 2018). This uncertainty parametrization can be used to better establish and account for interpretation uncertainties on-top of deterministic modelling workflows using a hybrid approach based on a single deterministic or multiple deterministic models. For example, a deterministic fault network topology model with fault throw uncertainties parameterized, or multiple deterministic models to characterize the most probable fault network topologies (e.g. Fig. 5) with fault throw uncertainties parameterized. Such hybrid approaches may provide the best solutions when the time and computational costs of full stochastic modelling are too high and/or when elements of the uncertainty in the geomodel are not best represented as simple stochastic functions, such as different conceptual models (e.g. for fault network topologies). Stamm et al. (2019) have recently explored how both fault throw and fault sealing uncertainty can be incorporated into stochastic geological modeling workflows, and studies like ours can help inform stochastic parametrization with how fault throw uncertainties can change along strike depending on changes in the seismic data quality.

The work of Pakyuz-Charrier et al. (2018) discusses the importance of proper parametrization of input data measurement uncertainty when constructing stochastic geomodels, but little is known about the uncertainties in interpreting the dense 3-D seismic datasets to obtain such input data for structural geomodels. Our work not only provides a first look at how significant these uncertainties can be, but additionally provides a first order approximation for parametrization of fault and horizon interpretation uncertainties within stochastic geomodels based on seismic image quality surrounding fault interpretations. Future

research into how our findings could be integrated with the seismic expression of fault zones (e.g. Botter et al., 2014; Iacopini et al., 2016) could further our ability to parametrize stochastic geomodels directly from seismic data.

While student interpretations will most certainly reside within the upper range of interpretation uncertainty, but we argue that it nevertheless provides significant value to stochastic parametrization, especially to the emergent Bayesian approaches to stochastic structural geomodelling (Caers, 2011; de la Varga and Wellmann, 2016; Wellmann et al., 2017), as it could provide informed—but not overly constrained—prior parametrization that can be reduced by case-specific geological likelihood functions and auxiliary data integration. We also agree with Caers (2018) on the need for a rigorous methodology of falsification in geomodelling. The integration of adequately parameterized seismic interpretation uncertainties into a Bayesian geomodelling framework could enable a quality control of the interpretation by probabilistic assessment of the stochastic geomodel against geological likelihood functions (e.g. fault length and throw relationships, fault population distributions, analog studies).

Our findings underline the complexity involved in the adequate parametrization of interpretation uncertainty in stochastic geomodelling: while Normal distributions may capture uncertainty adequately in areas of good seismic imaging (Fig. 8a), skewed fat-tailed distributions (e.g. Cauchy; Fig. 8b) or even Uniform distributions (Fig. 8c) are reasonable choices with degrading seismic quality. When implicitly modelling 3-D geological surfaces, many approaches are based on both surface points and strike and dip information. The latter carry significantly higher amounts of information (Calcagno et al., 2008; Laurent et al., 2016; Grose et al., 2017) than the surface points, thus emphasizing the need to quantify their uncertainty if it is used to generate and constrain 3-D stochastic geomodels. The use of van Mises-Fisher distributions to model uncertainty of orientation vectors was shown as a robust way to describe surface orientation uncertainty (Pakyuz-Charrier et al., 2018), and our analysis could provide valuable information for their parametrization in areas of high interpretation uncertainty within sedimentary basins.

## 4.4 Implications for machine learning

Recent efforts in automating seismic interpretation through the use of, mainly, Neural Networks (NN) has been increasingly successful in interpreting high quality (synthetic) seismic data (e.g. Huang et al., 2017; Dramsch and Lüthje, 2018; Wu et al., 2018). But NNs inherently do not take into account any geological reasoning skills that could make sense of areas of low seismic image data quality, but rather infer abstract features from their training data. And while NNs can be constructed and trained probabilistically, and thus enable uncertainty quantification of their outputs, they are likely to require additional information about the structures to be interpreted in areas where interpretation suffers from low data quality. In our analysis of the uncertainties in interpretations of the Gullfaks field 3-D seismic cube the interpretations of faults in areas of low seismic image quality conform to known fault geometries and topologies and are likely informed by interpretations of adjacent higher quality seismic image data, rather than their uncertainty distributions being simply correlated to seismic image quality (Fig. 8). Such evidence highlights the nuances in geological interpretation, and the difficulties in creating algorithms that represent the complexities of human thought processes. Irrespective of how easy it is, or otherwise, to apply our findings to inform automated interpretation efforts, there is value in studies such as the one presented here in generating understanding as to when and where such machine learning processes maybe applied effectively and how they could be improved by integrating

geological knowledge, maybe in the form of *logic rules* that influence NN weights and biases (Hu et al., 2016). Lu et al. (2018) show the use of Generative Adversarial Networks (GANs) in improving fault interpretation of low-resolution seismic image data by generating supersampled high-quality seismic images from lower quality data. One possibility could be to train similar NNs on the structural geology represented by ideal synthetic seismic image data and artificially noisy perturbations of the same to let NNs learn how ideal structures that underlie noisy seismic image data might look like. They could then possibly generate possible higher-quality realizations of noisy, uncertain areas of seismic images to support interpretation efforts. Overall, the complex interplay of the underlying geology, computational and conceptual challenges of the machine learning approaches and the human-induced uncertainties will require strongly interdisciplinary approaches to combine state-of-the-art algorithms and geological domain knowledge to further automate the laborious and uncertain task of 3-D seismic interpretation.

## 5  Summary

Our study provides a first look and quantification of the scale of uncertainties involved in the structural interpretation of data-dense 3-D seismic data. We have found that:

- Fault placement uncertainty shows strong dependency on seismic data quality. The use of a seismic attribute (RMS Amplitude) showed promising first results to be used as a proxy for estimating fault interpretation uncertainties. This can be especially valuable for the parametrization of stochastic geomodels based on single seismic reference interpretations.

- The common use of Normal distributions as perturbance distributions in stochastic geomodelling seems inadequate in areas where interpretation uncertainty is high (low seismic data quality). Instead, uncertainty parametrization should always be directly linked to the surrounding seismic data quality and the use of near-uniform distributions can be recommended in areas of extremely poor interpretability.

- We found that relative trends of fault placement uncertainty can be approximated linearly with depth, although we recommend more research to further investigate the influence of seismic data quality in combination with changes in reflector continuity with depth.

- The student interpretations displayed significant uncertainties in fault network topologies—despite the information-dense 3-D seismic data—which can have critical impacts on decision making based on geomodels constructed from seismic interpretations.

Additional interpretation uncertainty studies on different seismic data are recommended to further our knowledge about how interpretation uncertainties depend on seismic image data quality. There is also an element of the individual, and an individuals prior knowledge, in interpretation and further work should consider how different cohorts behave when faced with interpretation challenges such as that posed here. These cohorts could define levels of expertise, tools or techniques used and training background, in an attempt to consider the influence of these factors on the statistical distributions of interpretation outcome. Such factors may also vary depending on different tectonic and stratigraphic settings, so inclusion of a range of

geological contexts may also be important. Together such uncertainties could be integrated into future geomodelling efforts and decision making, as well as informing training, workflows and expertise development.

*Code availability.* The Python library *uninterp* was written to provide convenient functionality for the wrangling of exported Petrel interpretation data and uncertainty analysis of multiple interpretations. It is a free, open-source library licensed under the GNU Lesser General Public License v3.0 (GPLv3.0). It is hosted as a git repository https://github.com/pytzcarraldo/uninterp on GitHub (DOI: 10.5281/zenodo.2593587).

*Competing interests.* The authors declare that they have no conflict of interest.

*Disclaimer.* This research was conducted within the scope of a Total GRC-funded postgraduate research project.

*Acknowledgements.* We would like to acknowledge the help of David Iacopini for providing access to the interpretation projects. We also thank both Adam Cawood and Rebecca Robertson for providing extensive information about the interpretation process and helpful input during the data analysis. Alexander Schaaf was supported by Total GRC UK research funding, with Clare E. Bond supported by a Royal Society of Edinburgh research sabbatical grant. Special thanks go to both Billy Andrews and Awad Bilal for their detailed reviews that helped us to significantly improve the manuscript of this work.

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
