# Peer review of "Quantification of uncertainty in 3-D seismic interpretation: implications for deterministic and stochastic geomodelling and"

_Solid Earth, 2019_

## Referee Comment (RC1) · Billy Andrews (Referee) · 4 Apr 2019

General comments:

This manuscript tackles a highly topical and important concept of interpretation uncertainty, with a particular focus on 3D seismic. An extensive and appropriate data set is presented which comprises 78 seismic interpretations of the Gullfaks field undertaken by undergraduate students. Interpretational differences in horizon and fault stick picks are reported and quantified, with variability attributed primarily to changes in seismic

data quality. The authors go on to discuss how these results have implications for deterministic and stochastic geo-modelling and machine learning.

Whilst the results are well presented, there are occurrences where a stronger analytical framework would lead to greater confidence in the results and improve the scientific rigor of the manuscript. I suggest the authors include an additional section to the discussion to further elaborate on the reasons behind the results, prior to discussing the multiple implications. The addition of a conclusion section would also give readers a brief overview of further work and the manuscripts key points, along with providing a clear take home message. Overall, I suggest that the paper would be of great interest to the readership of Solid Earth and should be accepted if the suggested revisions are implemented. I very much look forward to seeing the revised version of this manuscript and believe this contribution has the potential to become an important contribution to field of 3D seismic interpretation.

Major comments 1. Framework for results and use of mean/SD/median: The authors provide an extensive and interesting suite of student interpretations, however, a primary concern whilst reviewing the manuscript was the framework and analysis used to describe the results. My concerns can be split into two: (a) the statistical analyse the data and (b) the language and framework used when describing these results.

(a) Throughout the manuscript the authors use a combination of mean, standard deviation, median and IQR, however it is not always clear how the results are distributed. Standard deviation is often used as a measure of uncertainty, however, I have two potential issues with this. The first being that in some cases the results don't appear to be normally distributed (e.g. interpretation density in Fig 9). In the case of a skewed distribution standard deviation should be avoided, and I advise the use of Inter Quartile Range (IQR) instead. It would also be useful to see what the maximum uncertainty, and not just the IQR or standard deviation, with minimum/maximum values reported. My second issues with the use of standard deviation is that a specific standard deviation value could represent either a high or low value of uncertainty depending on the

mean at that point. For example, in your fault throw data for F3 (Fig 7b) the standard deviation remains ∼30 in the north of the profile. The median throws in this section however ranges from ∼50 to ∼120, meaning the level on uncertainty is considerably higher at a smaller throw. I suggest to give the reader more confidence, and to better understand the risk in different areas that the following should be considered when analysing the results: i. The shape of the distribution? Is the distribution the same along the whole profile? ii. If the distribution is not normal, then consider using median and IQR to describe the results. iii. The normalisation of results would aid the comparison of uncertainty; this should either be through the use of coefficient of variance, or in the case of non-normally distributed data then a quartile-based coefficient of variance (IQR/median).

(b) Throughout the manuscript there are several instances where the authors use subjective language when describing the results (e.g. 'significantly reduced', 'observe large', 'roughly correspond'). The description of the results would be improved through the quantification, or further description, of what the authors mean. It would also give readers more confidence in the results. In addition to this there are occurrences where statements such as 'a few students' are used (e.g. Page 7 Line 7). In these cases, the authors should be explicit in what a few means '7 students'.

2. Discussion section 'key findings' This paper provides an impressive analysis of a 1st pass data set aiming to quantify uncertainty in 3-D seismic interpretation, however, I felt the authors needed further elaboration on the reasons behind the presented results. The manuscript would be improved through reducing the key findings to a set of points (potentially even using bullet points), and adding a section which investigates the factors behind the presented uncertainty. From your results this appears to be split into two 'themes':

(a) Human factors: Several sections allude to how students build a mental model during interpretation (e.g. taking information from outside areas of degraded seismic image quality to inform decisions). It is somewhat lacking that the current literature in this

area is missing, and that this aspect remains unexplored. Do those who use outside trends to inform areas of poor coverage end up with a better interpretation? Another important comment you raise can be found on Page 13 line 5-7 where you allude to the order students undertook the analysis. This is another important aspect to consider and should be explored further, possibly referencing other work which shows people can vary interpretations through time (e.g. Scheiber et al., 2015 for lineament extraction). The role of human factors on the collection of 3D seismic data should be further explored.

(b) Technical factors: These are highlighted strongly in that uncertainty is higher in areas of lower seismic resolution. Seismic resolution will always be lower surrounding fault, due to the increased amount minor structures and local deformation, and such we can expect uncertainty to remain high in such areas. We know from other aspects of fault science that 'intersection zones' or larger offset faults tend to have a wider zone of damage, and hence zone of reduced seismic image quality. Can we use some of this information to aid the assignment of risk in these areas?

I was often left asking 'why is this the case?' and the answers weren't forthcoming in the discussion. Although I have provided examples of what I feel should be expanded upon, there where sections which could also be expanded and linked to published literature. If this section is added, some of the implication sections could be slightly scaled back in particular section 4.4 which I feel has the least direct link to your results.

3. Conclusions: I felt this manuscript really lacked was a clear finale. The authors present an extensive set of results, which have clear implications to the interpretation of 3D seismic, however, in my opinion fail to leave the reader with a clear take home message. This point links to the previous regarding the discussion and believe the discussion should be slightly restructured as above and a set of conclusions included which pulls together the findings, highlights the clear importance of these results (including beyond 3D seismic interpretation e.g. modelling from other sources) and raise future research directions. This would tie an important contribution together, and provide readers with a clear take home message.

4. Figure quality/readability: Some of this may be due to the uploaded PDF, however, I found several figures difficult to read and often containing areas where text was too small. Some examples include line-weights of sub-sections, text size of longitude/latitude and labels within panels and occasionally the chosen color scheme used was difficult to read either on the screen or when printed off. Detailed points are raised in specific comments.

Specific comments [page (line)] Abstract: Likely to need minor edits following the suggested edits to the manuscript.

1 (17-21): The introductory paragraph of the MS should be expanded to further define uncertainty. Conceptual uncertainty is first stated to be important in the 2nd paragraph, however, non-specialised readers would benefit from an explicit introduction to the different types of uncertainty (e.g. Bond, 2015; Tennert, 2007) and potentially how this effects the mental model of the interpreter.

2 (31-33): I suggest you make it clear that the study focuses on an interpretation boundary of a student exercise here, it currently sounds like it focuses on an area of a larger data set.

3 (Fig1): (a) Colour and line weight for section line and interpretation box is unclear, both in colour and in B&W. I advise a change in colour and that the line weight is increased. The text size in the insert to this panel is far too small, as is the longitude and latitude numbers. The addition of a scale bar to this panel would also aid the reader. (b) A scale should be added to this panel. (c) The formation names are poor quality in the uploaded PDF, and also slightly on the small side.

3 (2-3): Many questions come to mind with respect to the level of experience of the interpreters and in part the limitations of your dataset, which includes undergraduate students only. Some of these include: Did everyone have the same level of training?

What was the 'specialisms' in the sample set (i.e. how much seismic interpretation, structural geology, stratigraphy etc. was covered and was this equal in the students)? Also how long was spent by each student (If you have this data it would be interesting to see if those who spent more time interpreted differently to those who did not)? How comfortable where the students with using petrel & integrating well and seismic data?

3 (5): How much assistance was given in this? What was the variability in the interpreted horizon when assistance was given and how does this compare to the Top Ness. Can the difference between the Top Cretaceous and Base Cretaceous/Top Ness horizons show the effect of training in reducing uncertainty? Also if there is little variability in the Top Cretaceous, due to the supervision, will this not effectively 'pin' one end of the fault sticks to a lower range of displacements, effetely adding to the increase in U/C with depth attributed to a degradation of seismic image quality (I agree image quality decreasing with depth will also be a factor).

3 (7-8): Was there any difference in interpretation from students who used these different methods? How often was seeded tracking or manual interpretation used?

4 (Fig2): Increase the text size on the axis for clarity.

4 (2-3): Suggest the text about 90 interpretations be removed as is only mentioned here, and does not seem required.

4 (10): This is an impressive data set, however, I would be interested to see how this is spread between the students. I suspect, and you allude to on page 4 line 15, that the number of fault sticks interpreted varies extensively between students, and that this is an important aspect of uncertainty. This could also then be further analysed to see if there is a correlation between number of fault sticks and level of uncertainty.

4 (15): How is interpretation density defined?

5 (Fig3): I wonder how Fault 1 and Fault 2 are defined in the northern interpretation bin once they are merged.

[Figure]

5 (13): I worry that this is effected not only on the placement, but also on how many fault-sticks each student included. In areas of relatively certain offsets, which will likely be increased by the image quality, I would imagine more sticks will be chosen, thus increasing the apparent 'certianty' of the result.

6 (Fig4): I find the addition of the mean fault plane & k-values from Fossen and Hesthammer (1998) confusing as is, however, it is an important point which you make on Pg 11 ln 31-32. It would be made clearer to the reader if this mismatch was raised in the results, and later discussed in the 'Key findings'. A reminder that stereonet plots go from N to S actually on the figure and not just in the figure caption would also be helpful in this figure.

6 (6-7): I am struggling to pull three clusters out of the stereonet data presented in fig 4a, and instead can only see two. I agree the data should be split into three due to the sinusoidal shape based in the geographic location, however, this information is instead better portrayed in Fig 3a. I advise you reword accordingly.

7 (Fig5): I would like to know the skewness of the distributions, particularly if this changes down dip, this will impact how valid the use of standard deviation is (See major comment 1). I also wonder which fault show the most variability with depth and why. Comparing using either a coefficient of variance (if distributions are normal) or quartile based coefficient of variance could pull out more trends between the faults. Also although standard deviation increases with depth, how well the data fits the regression line seems to decrease, particularly for F1. For F2 and F3, and to some extent F1 there seems specific horizons which show increased/decreased spread which is not in agreement with the linear regression. Is there an underlying control here? (e.g. stratigraphic layer with good/poor seismic response?). Visually I would consider changing the 'picks above BCU...' from light grey as it is difficult to see, the regression lines for F2 and F3 are also unclear when viewed on the screen (fine when printed).

7 (7): How many students did this? This is a source of error/uncertainty and I feel

it should not be dismissed. What training/geological information was provided to the students and from this should they have factored in the 'geological unreasonableness' of the interpretations?

7 (12): I question why probability is quoted here, you have 78 interpretations, so feel that the numbers represent the total number of students who interpreted that network.

7(14): I feel this needs to be linked back to interpretation and not to 'probable'. Probable suggests that if 100 random people where to be selected then X% would choose option Y, which I think is misleading as there are more human factors involved here. I also feel it is prudent to describe in the MS the level of exposure students has with 'complex' fault topologies.

8 (Fig. 6): In part (a) I would advise that the y-axis is changed to # of students and not a percentage (see comment #). In part (b) I wonder how statistically different A & C are in the students data? Is there a distinct gap? (as topologically they are the same, and geometrically similar).

8 (6): How do you define 'relatively constant' uncertainty? How is it measured? See major comment 1.

9 (Fig 7): This figure makes a very important point, that uncertainty can vary spatially, however, a number of questions are raised in how the results are presented. My main concern is the use of median and standard deviation (Again see Major comment 1). Why is median used? If it is because the distribution is skewed, which I suspect it is, then it is not statistically robust to use standard deviation. I would also like to see the min and max values here (aka what is the maximum risk in this data set?). I suggest redrafting to either show standard deviation surrounding the mean, with min and max values displayed, or to show the IQR around the median again with min/max values. I prefer the second method and suspect similar trends would be observed. Visually I would consider increasing the text size of the annotations. Is standard deviation in any way related to throw? A +/- 30 meters on a 120 m offset fault is much better (25%)

than on a 50 m offset fault (60%) Is quoting exact values the best way to compare uncertainty?

10 (9): How many students interpreted the fault further to the East?

10 (11) to 11 (7): This section suffers from a lack of statistical analysis, a framework to describe these results would increase the rigor of this section. The data shows some very important trends, probably the most important point of the manuscript, and with a more robust statistical analysis the reader would have more confidence in the results and following discussion.

11 (10-18): You open this paragraph with a statement that you show that u/c is correlated to seismic reflector strength, then backtrack on line 13 to discuss human factors. I would suggest that either this paragraph is split and both sections elaborated, or that the topic sentence incorporates both concepts. See Major comment 2

11 (26): How strong is the Top Ness horizon? Does this effect how well it is interpreted?

11(31-32): How did Fossen and Hesthammer (1998) get their pole? What was there scale of observation (i.e. did they have the data to extrapolate the sinusoidal shape)? The work on this should be included in this part of the discussion.

12 (Fig 9): How do you define interpretation density; units should be added if applicable? Visually this figure could do with a general text size increase, with many areas of text being too small. I would also suggest a change of colour for the boxes in part (1).

13 (5-7): This is a potentially important point and raises a very important question 'what order did students interpret the cube?' If students are spending more time on a certain area, where data is of better quality then there are more factors to consider in why your results are different. Also does the style of interpretation change with time? I advise either that the key findings section be reduced to a summary (e.g. set of bullet points) and separate section added to explore the reasons behind the uncertainty, probably split into 'technical' (e.g. image quality) and 'human' (e.g. different mental models) and

that appropriate literature be added to this discussion.

14 (27-29): I think it would be unwise to suggest normal distributions, even in areas of good seismic data. I suspect in nearly all cases the distributions will be skewed. Most faults display an asymmetric damage zone, and such will also show an asymmetric signature in seismic, should the flat tail be towards the hanging wall?

15 (3): I found this an underwhelming end to a really neat data set. Although the implications for machine learning are indeed relevant, I feel the MS is crying out for a conclusion section which ties the findings together and includes the 'next stages' in tackling uncertainty in 3D seismic interpretation. The section itself also seems somewhat out of the remit of this work, and could conceivably either be reduced or cut to make space for a discussion into the reasons behind the results as suggested previously.

Please find additional minor comments/suggested text edits on the attached MS (many of which are included in the specific comments).

Please also note the supplement to this comment:
https://www.solid-earth-discuss.net/se-2019-54/se-2019-54-RC1-supplement.zip

---

## Referee Comment (RC2) · Awad Bilal (Referee) · 3 May 2019

General comments:

The manuscript by Schaaf and Bond 2019 describes an interesting and important concept of 3D seismic interpretation uncertainty. The authors analysed an extensive data set, comprising 78 seismic interpretations carried out by final year undergraduate students of a 3D seismic dataset from the Gullfaks Field, northern North Sea. Uncertainties related to fault and horizon interpretations are explored and quantified using

[Figure]

Python tools. The results of the research are then discussed with implications for deterministic and stochastic geomodelling as well as machine learning.

Overall, I believe the paper would be of great interest for the readers, and it could be accpted but with strong modifications. There are some instances where the scientific approach was oversimplified and didn't actually consider other controlling factors. I appreciate the general discussion done but I believe that the 'Key finding' section of the discussing part could be significantly modified and discussed in more depth. The manuscript also lacks a clear and concise summary or conclusions. The above general comments are discussed in further detail below and/or in the attached PDF file of the manuscript with my comments highlighted. For revision, please use both files. I look forward to reading the revised final version of manuscript.

Major comments:

1. While reviewing the manuscript, there have been some instances where the scientific approach was either oversimplified or need further clarifications. For example,

(a) The authors attributed uncertainty in fault and horizon interpretations to mainly image quality. However, it was not clear to me what does seismic quality means in the manuscript (i.e. low amplitude and homogenous reflectors? low amplitude, chaotic reflectors? or noisy seismic. etc.). In addition, there was no information about the seismic data in the materials and methods section. I appreciate the work done in section 3.3 (seismic data quality) but I worry about the use of RMS attribute (reflector strength) to assess seismic data quality. Strong seismic reflections do not necessary means high seismic quality. Clear definition of what the authors consider as high or poor seismic quality is therefore crucial. I suggest the authors start by describing the seismic stratigraphy of the sequences and the seismic characteristics of the analysed Top Ness horizon. Areas of poor seismic quality can then be highlighted together with the possible reasons behind low quality data (e.g. depositional environment, inherited low quality i.e. processing... etc). In order to give the reader more confidence, I

suggest including a new figure (as Fig. 2 or 3), comprising two, N-S and E-W seismic sections, showing the interpreted horizons and seismic quality.

(b) The authors observed that uncertainties in the horizon (Top Ness; Fig. 8) interpretation are significantly reduced surrounding wells with a general trend of increasing uncertainties from west to east. I suggest that this is further analysed and discussed. Important aspects to be considered are; (1) what is the seismic characteristics of the interpreted horizon? (2) what is the structural configuration (faulted, non-faulted) of the horizon; (3) how confident the students/authors are with both formation tops and well locations; and (4) is there any uncertainty in formation tops that can be compared/correlated to the uncertainties analysed in this paper. Additional factors that might also have affected interpretation include the level of assistance to students, time spent in the interpretation, the methodology and order of interpretation and how familiar the students were with using Petrel.

The fact that uncertainty increases away from well location is a general statement and, in many cases, it is invalid unless other factors are considered. Horizon interpretation uncertainty away from well location largely depends on the continuity of the horizon (also dependent on the depositional environment) and structural complexities. There are many cases where uncertainty is significantly low for hundreds of kilometres away from well location. Other cases show high uncertainty in near to the well or even in the Formation tops of the well.

(c) I worry about the uncertainty of fault throw. For example, Figure 7b. shows that uncertainty is higher at smaller median throw (southern part of the fault), while it is expected to see high uncertainty at larger throw value. The figure also shows that same amount of median throw can have both high and low uncertainties (see central part of the fault ∼Y 6784000). This can also be observed in Fig. 7a. Generally, uncertainty in throw mainly depends on the uncertainty of the correlation of the stratigraphic markers in the hangingwall and footwall blocks of the fault. For example, uncertainty in fault throw will be low where the stratigraphic correlation between fault blocks is good

independently of throw value. In more detail, the uncertainty in cut-off angle in each sides of the fault depends on many aspects such as structural complexity, fault plane definition on the seismic, seismic quality near faults. The new publication of Ze and Alves titled 'Impacts of data sampling on the interpretation of normal fault propagation and segment linkage' might be of interest.

2. I believe that the manuscript would be greatly improved and give more confidence to the reader if the 'Key findings' section of the discussion part is discussed in more depth. The fact that uncertainty in seismic interpretation increases in areas of poor seismic quality and linkage zones is relatively well known and expected. I appreciate the work done on quantifying these uncertainties, but I advise that the results of the paper are discussed in more detail. One important aspect that could be highlighted is the reasoning behind the variability in seismic interpretations, taking into consideration other factors that might affect seismic interpretation. Human bias in seismic interpretation and the expertise of the interpreters can also be addressed. It would also be interesting to see whether the result would change if the interpretation was carried out by experienced seismic interpreters using seismic filters to improve seismic quality together with various attributes (i.e. the effect of the level of experience). The authors can also analyse the effect of training (supervised top Cretaceous horizon interpretation) in reducing horizon interpretation uncertainty.

3. While reviewing the manuscript, there have been several instances where the description was not clear, or the authors use a great deal of qualitative language. I would prefer some quantifications of what few, major etc mean. These together with other minor corrections are highlighted in the attached original manuscript.

4. Conclusions: This paper describes a very important aspect of 3D seismic interpretation. However, a clear and concise summary with a clear link to the results and discussion would highlight the significance of the paper and results. Hence, I suggest the authors add a short summary or conclusions as bullet points with a clear take-home message.

5. Figures: Overall, the figures of the manuscript are clear and well-presented but please make sure of the resolution in the final printed version. Text size in some figures needs to be increased. Several figures have neither a scale bar nor north arrow. I also have suggested re-drafting and/or adding three new figures. Detailed comments on these new figures and on each figure are highlighted in the attached original manuscript.

Detailed comments:

P1 – Abstract:

Overall, it provides a clear summary but I believe it will be updated according to the reviewer's comments.

1. Introduction:

Overall, the text is clear but I suggest the authors further clarify uncertainty in seismic interpretation and make it accessible to non-specialists. The authors could simply address that conceptual uncertainty mainly comes from human bias in seismic interpretation and depends on the expertise of the interpreter. Also, other types of uncertainties such as geophysical uncertainty, uncertainties related to formation tops and check-shot data could also be highlighted.

L12 – Please add the name of the 3D seismic survey

L14 – I would be more specific here. Seismic noise at depth? Near faults?

L19 – The authors have shown a simple fault interpretation of Fossen and Hesthammer (1998) as a reference example. How about horizon interpretation? Do you have an interpreted horizon? Please find further comments on this below (i.e. Fig. 8).

P2 – Section 2.1 Gullfakes geology and seismic data:

The paragraph lacks information about the 3D seismic data. Key information includes (1) the name of the 3D seismic survey; (2) acquisition year; (3) whether the data is

originally in depth or time? and (4) key information about processing history as this will further clarify the seismic quality issue. I would also add the key tectonic phases evolution of the area (i.e. how many phases and ages).

L30-33 – Please clarify the research focus area both here and in Figure (1). I also suggest clarifying the scale of the interpretation area. Could you also highlight that the overall structure of the area is very complex particularly in the accommodation zone and collapsed footwall structures in the horst complex zone, while the western domino-system (research focus) is comparatively simple.

P3_Figure 1 - Please increase text size particularly in Fig. 1a and the Formation names in Fig. 1c. The location of the 3D seismic is not clear. The schematic fault map (b) is oversimplified. Please show fault displacement (fault polygons) and dip direction. Fault polygons would greatly help the reader to understand linkage zones as well as major and minor faults. I wonder how Fault 1 was identified in the northern part where it merges with Fault 2. Please see further comments highlighted on the attached original manuscript.

P3 – Section 2.2 Interpretation dataset:

Many questions came to my mind with respect to seismic stratigraphy. What are the seismic characteristics of the Top Ness horizon and why it was selected for analysis? Please add this key information to give greater confidence to the reader with respect to the results.

P3_L4 – Please show the location of the wells in Figure 1. How confident you are with well locations and Formation tops.

P3_L8 – Why did students use manual interpretation? how often seeded tracking and manual interpretation were used?

P3_L9 – Do you mean interpolated auto-tracked surfaces? Did students actually use gridding? I would be more specific

P4_L3 – Why the Top Ness horizon was selected for analysis. Why not BCU

P4_Figure 2 - Please increase figure and text size. Also show both North arrow and a scale bar.

P4 – Section 2.3 Data analysis:

It was not clear to me the key steps of the analysis. I would suggest adding a diagram showing steps of the workflow applied as this would greatly help the reader in particularly non-specialists. Please see further minor comments highlighted on the attached original manuscript.

P5_L8 – I am not aware of RMSA. I am sure RMS only (without 'A') would be fine.

P5_6 - Figures 3 & 4 – I would suggest merging these figures or having both of them in the same page to make it easy for the reader to follow. Please make sure of the quality of the figure in the final printed version of the paper.

P6_L2-3 – What do you mean by spread? Can you make this clear?

P7_L13-14 – Why F3 was omitted? If it was interpreted in a similar fashion, why does uncertainty increases southwards as shown in Figure 3c?

P8_Figure 6 – Can you show students number in the Y-axis of Fig. 6a. Please add fault numbers (e.g. F1, F2) and dip direction in Fig. 6b. I was wondering how the authors differentiated between A and C and how fault 1 was defined in the northern part where it merges with fault 2 (Fig. 6b).

P8 – Section 3.2 Fault throw and horizon uncertainty:

It was not clear to me whether the throw analysis was carried out along fault strike or a strike-parallel window.

What are the reasons behind selecting faults 1 and 3 (which was interpreted in a similar fashion and omitted in the previous section) for throw analysis?

P9_Figure 7a. - How about uncertainty in the linkage zone? I think that the increase in uncertainty near survey edge is more likely related to the complex linkage zone to the south where faults 1, 2a and 2b merge as shown in Figure 1. It is well known that fault interaction can influence fault propagation and hence, displacement profiles. Do you think that the displacement profile shown in Fig. 7a would change if fault 2 was added to fault 1. It will be very interesting to see both faults.

P9_Figure 7b. – As indicated in the general comments above, this figure shows higher uncertainty associated with small median throw. Also, same amount of median throw show both high and low uncertainty. Please make of the analysis. Please see further comments highlighted in the attached original manuscript.

P9_L4-9 - One key question came to my mind while reading the last paragraph of page 9 (description of Figure 8) is that where is the actual interpretation of the Top Ness Formation. I strongly suggest that the best interpreted horizon is added. so the reader can see the actual interpretation of both horizon and faults. It would be even better if it was interpreted by experienced seismic interpreter.

P10_Figure 8 – Please redraft to include (a) the actual (best) interpreted horizon, (b) average horizon, and (c) the standard deviation horizon. Also, please add north arrow and scale bar in all parts of the figure. Please see further comments highlighted in the attached original manuscript.

P10 – Section 3.3 Seismic data quality:

As indicated in the general comments above. I worry about the title of this section as well as the framework and results. The strength of seismic reflector dose not necessarily mean good seismic quality. I also worry about the use of a time slice (Fig. 9.1 & 9.2) to assess seismic quality based on the strength of a single seismic reflector as the results can be significantly different at different levels or even the same level. I believe that interpretation of fault 3 was aided in the northern part of the fault (Fig. 9.1 & inline section A) because of both the sharp reflection termination and the good correlation between seismic reflection packages on both sides of the fault. Generally, fault interpretation dose not actually depends on individual reflections, instead packages of reflections across the fault. I would suggest the authors further test their results and see if the same approach can be applied to Fault 1 and 2 where the high amplitude to the south (Fig. 9) coincide with high uncertainty in both fault (Fig. 3) and horizon (Fig. 8) interpretation. Moreover, the authors could test whether the results are same if (1) the time slice was replaced by RMS attribute extraction on the actually interpreted horizon; or (2) a different time slice (e.g. $\sim$2100 or 2300) was used. Further minor comments on this and Figure 9 are highlighted in the attached original manuscript.

Please see the attached original manuscript for further minor comments on the discussion part.

Please also note the supplement to this comment:
https://www.solid-earth-discuss.net/se-2019-54/se-2019-54-RC2-supplement.zip

—————————————————————

---

## Author Comment (AC1) · 6 Jun 2019

Thank you very much for your helpful and in-depth review of our manuscript. We have thoroughly adapted our manuscript to incorporate the improvements you've raised. Mainly, we have focused on providing the reader with a more rigorous statistical presentation of our results and the addition of a summary section at the end of the manuscript. We have also made numerous changes to our figures to improve their quality and incorporated many of your minor comments provided by your supplementary material. Please find our detailed responses to your comments

in the supplementary material, along with the revised manuscript and change-tracked manuscript.

Please also note the supplement to this comment:
https://www.solid-earth-discuss.net/se-2019-54/se-2019-54-AC1-supplement.zip

———————————————

---

## Author Comment (AC2) · 6 Jun 2019

Thank you very much for your in-depth review of our manuscript. We have incorporated a lot of your input and believe that the paper has significantly improved in quality. We have added a definition of what we define as seismic data quality within the context of seismic interpretation, significantly improved the Figures according to your recommendation and have added all information we have about the seismic dataset. We have also added a summary section to the end of the manuscript to provide the reader with a clear take-away message.

[Figure]

Please find all of our detailed responses to your comments in the supplementary material, along with both the revised and change-tracked manuscript.

Please also note the supplement to this comment:
https://www.solid-earth-discuss.net/se-2019-54/se-2019-54-AC2-supplement.zip

---

## Author Response (AR1)

**Author Response to RC1 (Billy Andrews)**

**Major comments responses**

**[1]** The authors provide an extensive and interesting suite of student interpretations, however, a primary concern whilst reviewing the manuscript was the framework and analysis used to describe the results. My concerns can be split into two: (a) the statistical analyse the data and (b) the language and framework used when describing these results.

**[1a]** Throughout the manuscript the authors use a combination of mean, standard deviation, median and IQR, however it is not always clear how the results are distributed. Standard deviation is often used as a measure of uncertainty; however, I have two potential issues with this. The first being that in some cases the results don't appear to be normally distributed (e.g. interpretation density in Fig 9). In the case of a skewed distribution standard deviation should be avoided, and I advise the use of Inter Quartile Range (IQR) instead. It would also be useful to see what the maximum uncertainty, and not just the IQR or standard deviation, with minimum/maximum values reported. My second issues with the use of standard deviation is that a specific standard deviation value could represent either a high or low value of uncertainty depending on the mean at that point. For example, in your fault throw data for F3 (Fig 7b) the standard deviation remains ~30 in the north of the profile. The median throws in this section however ranges from ~50 to ~120, meaning the level on uncertainty is considerably higher at a smaller throw. I suggest giving the reader more confidence, and to better understand the risk in different areas that the following should be considered when analysing the results: i. The shape of the distribution? Is the distribution the same along the whole profile? ii. If the distribution is not normal, then consider using median and IQR to describe the results. iii. The normalisation of results would aid the comparison of uncertainty; this should either be through the use of coefficient of variance, or in the case of non-normally distributed data then a quartile-based coefficient of variance (IQR/median).

**AUTHORS COMMENT**: Thank you for the suggestions! They enabled us to make the manuscript more robust with regards to the statistical description of our findings! We agree with the reviewer in that the use of IQR instead of standard deviation is a better way to analyse the fault throw uncertainty.

**CHANGE IN MANUSCRIPT**: We have redrafted Figure 7 (now 6) as boxplots of the fault throw uncertainty data and have adapted our description of the findings accordingly.

**[1b]** Throughout the manuscript there are several instances where the authors use subjective language when describing the results (e.g. 'significantly reduced', 'observe large', 'roughly correspond'). The description of the results would be improved through the quantification, or further description, of what the authors mean. It would also give readers more confidence in the results. In addition to this there are occurrences where statements such as 'a few students' are used (e.g. Page 7 Line 7). In these cases, the authors should be explicit in what a few means '7 students'.

**AUTHORS COMMENT**: Thank you for pointing this out! We have adapted the manuscript to remove the use of subjective language where possible. We have improved the statistical rigor in our description of results.

**CHANGE IN MANUSCRIPT**: A multitude of changes throughout the manuscript. Please see the revised manuscript or the change-tracked manuscript.

**[2] Discussion section 'key findings':** This paper provides an impressive analysis of a 1st pass data set aiming to quantify uncertainty in 3-D seismic interpretation, however, I felt the authors needed further elaboration on the reasons behind the presented results. The manuscript would be improved through reducing the key findings to a set of points (potentially even using bullet points), and adding a section which investigates the factors behind the presented uncertainty. From your results this appears to be split into two 'themes':

**AUTHORS COMMENT**: We have structured our discussion into four parts to first present to the reader the key findings of our study, and then to address the implications on what we believe to be the most important applications of seismic interpretation. We have added a summary section though, to provide the reader with a clear take-home message following our discussion.

**CHANGE IN MANUSCRIPT**: We have done various changes throughout our discussion, please view the revised or change-tracked manuscript.

**[2a] Human factors**: Several sections allude to how students build a mental model during interpretation (e.g. taking information from outside areas of degraded seismic image quality to inform decisions). It is somewhat lacking that the current literature in this area is missing, and that this aspect remains unexplored. Do those who use outside trends to inform areas of poor coverage end up with a better interpretation? Another important comment you raise can be found on Page 13 line 5-7 where you allude to the order students undertook the analysis. This is another important aspect to consider and should be explored further, possibly referencing other work which shows people can vary interpretations through time (e.g. Scheiber et al., 2015 for lineament extraction). The role of human factors on the collection of 3D seismic data should be further explored.

**AUTHORS COMMENT**: We have no real way of telling which students used outside trends to inform their interpretation in areas of poor coverage. We also have no information about the exact order students interpreted specific faults or fault blocks, as this would have required capturing and analysing their screen throughout the entire interpretation process. We therefore try to abstain from speculation about psychological effects for which we have no evidence for the dataset. We think these factors are vitally important to understanding all sources of interpretation uncertainty, but our manuscript is concerned with quantifying the effect size of this uncertainty and connect it to seismic data quality.

**CHANGE IN MANUSCRIPT**: -

**[2b] Technical factors**: These are highlighted strongly in that uncertainty is higher in areas of lower seismic resolution. Seismic resolution will always be lower surrounding fault, due to the increased amount minor structures and local deformation, and such we can expect uncertainty to remain high in such areas. We know from other aspects of fault science that 'intersection zones' or larger offset faults tend to have a wider zone of damage, and hence zone of reduced seismic image quality. Can we use some of this information to aid the assignment of risk in these areas?

I was often left asking 'why is this the case?' and the answers weren't forthcoming in the discussion. Although I have provided examples of what I feel should be expanded upon, there where sections which could also be expanded and linked to published literature. If this section is added, some of the implication sections could be slightly scaled back in particular section 4.4 which I feel has the least direct link to your results.

**AUTHORS COMMENT**: The assignment of risk or uncertainty based on fault offset is not straight forward in our opinion. Our study results have shown, that the seismic data quality around e.g. the

southern part of Fault 3 is low in combination with low throw. Seismic data quality is affected by so many aspects from the sedimentology and structural geology of the subsurface imaged to the imaging itself and the processing. Our study attempts to bypass exactly this to allow for estimation of uncertainties by using seismic data quality itself as a proxy for it (Fig. 8). Thus we believe the in-depth discussion of the multitude of impacting factors on seismic data quality out of scope for this manuscript, especially because it is very challenging research to identify the amount of impact every of the aforementioned aspects have.

**CHANGE IN MANUSCRIPT**: We have added references regarding the impact of fault zones on seismic data quality for readers to follow.

**[3] Conclusions**: I felt this manuscript really lacked was a clear finale. The authors present an extensive set of results, which have clear implications to the interpretation of 3D seismic, however, in my opinion fail to leave the reader with a clear take home message. This point links to the previous regarding the discussion and believe the discussion should be slightly restructured as above and a set of conclusions included which pulls together the findings, highlights the clear importance of these results (including beyond 3D seismic interpretation e.g. modelling from other sources) and raise future research directions. This would tie an important contribution together and provide readers with a clear take home message.

**AUTHORS COMMENT**: We have added a summary/conclusion section to the end of the paper to provide the readers with a clear tame home message

**CHANGE IN MANUSCRIPT**: Please see the revised manuscript for the Summary section.

**[4]** Figure quality/readability: Some of this may be due to the uploaded PDF, however, I found several figures difficult to read and often containing areas where text was too small. Some examples include line-weights of sub-sections, text size of longitude/ latitude and labels within panels and occasionally the chosen color scheme used was difficult to read either on the screen or when printed off. Detailed points are raised in specific comments.

**AUTHORS COMMENT**: Thank you for pointing this out. We have adapted the figures to improve text sizes and legibility of colours and line weights where applicable.

**CHANGE IN MANUSCRIPT**: Figure changes throughout the manuscript. Please see the revised manuscript.

**Detailed comments responses**

**1 - Introduction**

**[1 (17-21)]:** The introductory paragraph of the MS should be expanded to further define uncertainty. Conceptual uncertainty is first stated to be important in the 2nd paragraph, however, non-specialised readers would benefit from an explicit introduction to the different types of uncertainty (e.g. Bond, 2015; Tannert et al., 2007) and potentially how this effects the mental model of the interpreter.

**AUTHORS COMMENT:** We agree that a more specific description of the subjective uncertainty of interpretation and how it contrasts with objective measurement uncertainty would increase clarity of the manuscript.

**CHANGE IN MANUSCRIPT:** Our work is thus concerned with quantifying the scope of uncertainties in seismic interpretation, which represents inevitably biased, human judgment under uncertainty

(Tversky and Kahneman, 1974). This "subjective" uncertainty is in contrast to more "objective" uncertainty (Tannert et al., 2007; Bond, 2015) related to the geophysical acquisition of the data itself.

**2.1 – Gullfaks geology and seismic data**

**2 (31-33):** I suggest you make it clear that the study focuses on an interpretation boundary of a student exercise here, it currently sounds like it focuses on an area of a larger data set.

**AUTHORS COMMENT:** We agree, and improved clarity of what area is part of the study both in text and in Figure 1.

**CHANGE IN MANUSCRIPT:** Our study focuses on the structurally simpler western part of the domino system, where we investigate the uncertainty of the three faults F1-F3 and the fault blocks A-E depicted in Figure 1b.

**3 (Fig1):** (a) Colour and line weight for section line and interpretation box is unclear, both in colour and in B&W. I advise a change in colour and that the line weight is increased. The text size in the insert to this panel is far too small, as is the longitude and latitude numbers. The addition of a scale bar to this panel would also aid the reader. (b) A scale should be added to this panel. (c) The formation names are poor quality in the uploaded PDF, and also slightly on the small side.

**AUTHORS COMMENT**: We have adapted the color and line weights of annotations in Figure 1a to improve legibility.

**CHANGE IN MANUSCRIPT**: Figure changes.

**3 (2-3):** Many questions come to mind with respect to the level of experience of the interpreters and in part the limitations of your dataset, which includes undergraduate students only. Some of these include: Did everyone have the same level of training? What was the 'specialisms' in the sample set (i.e. how much seismic interpretation, structural geology, stratigraphy etc. was covered and was this equal in the students)? Also how long was spent by each student (If you have this data it would be interesting to see if those who spent more time interpreted differently to those who did not)? How comfortable where the students with using petrel & integrating well and seismic data?

**AUTHORS COMMENT:** The students all went through the undergraduate studies at Aberdeen, and this was their introduction course to 3-D seismic interpretation using Petrel. We do not have access to the time spent by each student, as they did significant amounts of the interpretation outside formal class times. We agree that this would be an interesting variable to analyse though! We have clarified the student's level of knowledge in seismic interpretation and Petrel.

**CHANGE IN MANUSCRIPT:** While the students had prior training in structural geology and interpretation of 2-D seismic data, this was the student's first hands-on course in 3-D seismic interpretation using the Petrel software as part of their undergraduate program.

**3 (5):** How much assistance was given in this? What was the variability in the interpreted horizon when assistance was given and how does this compare to the Top Ness. Can the difference between the Top Cretaceous and Base Cretaceous/Top Ness horizons show the effect of training in reducing uncertainty? Also, if there is little variability in the Top Cretaceous, due to the supervision, will this not effectively 'pin' one end of the fault sticks to a lower range of displacements, effetely adding to the increase in U/C with depth attributed to a degradation of seismic image quality (I agree image quality decreasing with depth will also be a factor).

**AUTHORS COMMENT:** Comparing the difference in interpretations between Top Cretaceous (TC) and both the Base Cretaceous Unconformity (BCU) and Top Ness horizon (TN) to assess the effect of training or supervision on interpretation uncertainty is unlikely to give clear information: The TC horizon is very easy to pick due to its lack of significant deformation and we believe it would not provide a valid comparison with the strongly deformed TN. The faults investigated do not reach above the BCU and thus do not affect the TC.

**CHANGE IN MANUSCRIPT**: -

**3 (7-8):** Was there any difference in interpretation from students who used these different methods? How often was seeded tracking or manual interpretation used?

AUTHORS COMMENT: We have no information about the specific interpretation tools used by students at any given time. It would make an interesting study though to assess bias introduced by software tools!

**CHANGE IN MANUSCRIPT:** -

**4 (Fig2):** Increase the text size on the axis for clarity.

AUTHORS COMMENT: We have increased the text size for axis labels / ticks.

CHANGE IN MANUSCRIPT: Changes in Figure 2.

**4 (2-3):** Suggest the text about 90 interpretations be removed as is only mentioned here and does not seem required.

**AUTHORS COMMENT**: While the information is not further discussed in the manuscript, we believe it relevant to mention any (subjective) prior filtering we have applied to the dataset.

**CHANGE IN MANUSCRIPT**: -

**2.3 – Data analysis**

**(10)**: This is an impressive data set; however, I would be interested to see how this is spread between the students. I suspect, and you allude to on page 4 line 15, that the number of fault sticks interpreted varies extensively between students, and that this is an important aspect of uncertainty. This could also then be further analysed to see if there is a correlation between number of fault sticks and level of uncertainty.

**AUTHORS COMMENT**: Thank you for pointing this out! We conducted a Bayesian estimation of the differences for the overall standard deviation of fault stick placement between the students falling into the categories of above and below 50% of fault stick interpretation frequency. We have added this to the results section.

**CHANGE IN MANUSCRIPT**:  Analysis of the effect of fault stick interpretation frequency between the students with below-median and above-median fault stick interpretation frequency on overall fault standard deviation was analysed using Bayesian estimation (Kruschke, 2013). We observed a difference in mean standard deviation of $35.8~m$, $20.2~m$ and $81.5~m$ for Fault 1, 2 and 3 respectively, with probability of differences being larger than zero being $99.3~\%$, $87.4~\%$ and $99.9~\%$, making the differences for Fault 1 and 3 statistically credible.

**4 (15):** How is interpretation density defined?

**AUTHORS COMMENT:** We have removed the possibly confusing wording of interpretation density (number of interpretations per volume) and just refer to it as frequency.

**CHANGE IN MANUSCRIPT:** In the following analysis we present 2-D and slices of 3-D histograms of fault interpretations, showing interpretation frequency across the domain.

**3 - Results**

**5 (Fig3):** I wonder how Fault 1 and Fault 2 are defined in the northern interpretation bin once they are merged.

**AUTHORS COMMENT:** This is dependent on the student interpretation. If they interpreted Fault 1 as continuing or Fault 2, respectively.

**CHANGE IN MANUSCRIPT:** -

**5 (13):** I worry that this is affected not only on the placement, but also on how many fault-sticks each student included. In areas of relatively certain offsets, which will likely be increased by the image quality, I would imagine more sticks will be chosen, thus increasing the apparent 'certainty' of the result.

**AUTHORS COMMENT**: Indeed, as this is a basic histogram of the fault sticks, high frequency interpretations can affect the plot. But these differences in frequency are along strike and should have a negligible effect on the fault placement uncertainty patterns observed orthogonal to strike.

**CHANGE IN MANUSCRIPT: -**

**6 (Fig4):** I find the addition of the mean fault plane & k-values from Fossen and Hesthammer (1998) confusing as is, however, it is an important point which you make on Pg 11 ln 31-32. It would be made clearer to the reader if this mismatch was raised in the results, and later discussed in the 'Key findings'. A reminder that stereonet plots go from N to S actually on the figure and not just in the figure caption would also be helpful in this figure.

**AUTHORS COMMENT**: We have added an explanation of the Fossen & Hesthammer (1998) to the main text in the results section to improve clarity. We have also merged Figure 3 and 4 into 3, which will enable the reader to identify more easily which Stereonet plot belongs into which bin.

**CHANGE IN MANUSCRIPT**: We have added Bingham mean poles from \citet{fossen_structural_1998} for all three faults in the plot (light blue) for comparison.

**6 (6-7):** I am struggling to pull three clusters out of the stereonet data presented in fig 4a, and instead can only see two. I agree the data should be split into three due to the sinusoidal shape based in the geographic location, however, this information is instead better portrayed in Fig 3a. I advise you reword accordingly.

**AUTHORS COMMENT**: The clusters correspond to the three bins. You can see the clusters when separated into the three separate Stereonet plots. This should be more clear to the reader now that Figure 3 and 4 were merged. We have updated the figure reference accordingly.

**CHANGE IN MANUSCRIPT**: Fault 1 shows three distinct clusters of orientations (Fig. 3A, a-d), [...]

**7 (Fig5):** I would like to know the skewness of the distributions, particularly if this changes down dip, this will impact how valid the use of standard deviation is (See major comment 1). I also wonder which fault show the most variability with depth and why. Comparing using either a coefficient of

variance (if distributions are normal) or quartile-based coefficient of variance could pull out more trends between the faults.

Also, although standard deviation increases with depth, how well the data fits the regression line seems to decrease, particularly for F1. For F2 and F3, and to some extent F1 there seems specific horizons which show increased/decreased spread which is not in agreement with the linear regression. Is there an underlying control here? (e.g. stratigraphic layer with good/poor seismic response?). Visually I would consider changing the 'picks above BCU. . .' from light grey as it is difficult to see, the regression lines for F2 and F3 are also unclear when viewed on the screen (fine when printed).

**AUTHORS COMMENT**: Median skewness of the distributions show approximate symmetry to moderate skewness of the distributions, but with significant standard deviation (see Table 1). We concluded from that, that the use of standard deviation is adequate for the purpose of this paper. We use here a basic categorization of -0.5 to 0.5 as approximately symmetric and ±0.5 to ±1.0 as moderately skewed. Overall, the variation of skewness with depth is quite high. We are aware that collapsing all the information into a single scatterplot is a significant simplification of the data, but we think this to be adequate for the first-look scope of the paper.

The quality of the seismic response of horizons is a major controlling factor of interpretation uncertainty and will certainly have a strong impact on uncertainty. As we have shown in Figure 9, strong bounding seismic reflectors significantly reduce interpretation uncertainty of faults. In this plot we collapse the entire fault uncertainty along the y-axis of the domain, thus averaging across a large area which makes the correlation with specific physical features challenging, and we did not pursue for this manuscript.

*Table 1: Median skewness for fault position with depth.*

|  | **Fault 1** | **Fault 2a** | **Fault 2b** | **Fault 3** |
|---|---|---|---|---|
| Median Skewness | -0.06 ± 0.36 | 0.25 ± 0.36 | -0.20 ± 0.33 | 0.21 ± 0.27 |

**CHANGE IN MANUSCRIPT**: Changed grey Figure 5 text to black for better legibility.

**7 (7):** How many students did this? This is a source of error/uncertainty and I feel it should not be dismissed. What training/geological information was provided to the students and from this should they have factored in the 'geological unreasonableness' of the interpretations?

**AUTHORS COMMENT:** We did not count the number of interpretations that do so, but removed any fault stick point above the mean BCU from the analysis automatically. But we believe that most students simply just did not terminate their fault interpretations correctly, as no geological model was built afterwards. We do not believe that most students who did this actually interpreted what they thought were meaningful faults above the BCU conceptually, but rather due to lack of attention to this detail, as this was not a focus of their coursework.

**CHANGE IN MANUSCRIPT:** -

**7 (12):** I question why probability is quoted here, you have 78 interpretations, so feel that the numbers represent the total number of students who interpreted that network.

**AUTHORS COMMENT:** Used student counts for clarity.

**CHANGE IN MANUSCRIPT:** Five modes of FN topology make up the bulk of fault network topologies, while others were only interpreted by 3 or less students respectively.

**7(14):** I feel this needs to be linked back to interpretation and not to 'probable'. Probable suggests that if 100 random people where to be selected then X% would choose option Y, which I think is misleading as there are more human factors involved here. I also feel it is prudent to describe in the MS the level of exposure students has with 'complex' fault topologies.

**AUTHORS COMMENT:** Changed probable to frequent for clarity.

**CHANGE IN MANUSCRIPT:** Note that the most frequent FN (Fig. 6b, A) is different from the reference expert FN interpretation …

**8 (Fig. 6):** In part (a) I would advise that the y-axis is changed to # of students and not a percentage (see comment #). In part (b) I wonder how statistically different A & C are in the students data? Is there a distinct gap? (as topologically they are the same, and geometrically similar).

**AUTHORS COMMENT**:  Added student numbers to the probability mass plot for clarity. We also enhanced the clarity of the related schematic fault networks.

**CHANGE IN MANUSCRIPT**: Figure 6 changed.

**8 (6):** How do you define 'relatively constant' uncertainty? How is it measured? See major comment 1.

**AUTHORS COMMENT:** We are referring to no apparent changes in the uncertainty pattern of the fault throw along strike. We clarified that in the manuscript.

**CHANGE IN MANUSCRIPT:** Fault throw uncertainty shows no apparent pattern change along the strike (around $\pm 30~m$), while rising sharply to about $\pm75~m$ at the southern edge of the seismic cube.

**9 (Fig 7):** This figure makes a very important point, that uncertainty can vary spatially, however, a number of questions are raised in how the results are presented. My main concern is the use of median and standard deviation (Again see Major comment 1). Why is median used? If it is because the distribution is skewed, which I suspect it is, then it is not statistically robust to use standard deviation. I would also like to see the min and max values here (aka what is the maximum risk in this data set?). I suggest redrafting to either show standard deviation surrounding the mean, with min and max values displayed, or to show the IQR around the median again with min/max values. I prefer the second method and suspect similar trends would be observed.

Visually I would consider increasing the text size of the annotations. Is standard deviation in any way related to throw? A +/- 30 meters on a 120 m offset fault is much better (25%) than on a 50 m offset fault (60%) Is quoting exact values the best way to compare uncertainty?

**AUTHORS COMMENT**: Thank you for pointing this out! We have adapted the plots to boxplots to show median fault throw with IQR, minima and maxima, as well as outliers to improve the robustness of our analysis. Overall the same trends visualize, but the difference in fault throw uncertainty at the merge of Fault 1 and 2 are much more visible. We have modified the results description accordingly.

We have increased font sizes in the figures.

**CHANGE IN MANUSCRIPT**: Results of the fault throw analysis are plotted in Fig. \ref{fig:07}. The boxplots show median fault throw with the associated interquartile range (IQR), extrema and outlier values along fault strike direction. The throw profile of Fault 1 (Fig. \ref{fig:07}a) shows a distinct sinuous shape spatially associated with its interaction with Fault 2. This shows one bin with high median fault throw of approx. $180~m$ and high fault throw uncertainty before strongly decreasing in fault throw values down to a median of about $40~m$ and one of the lowest IQR along the fault. Median fault throw then rises steadily towards the South while increasing in uncertainty. Notice the increase of uncertainty at both ends of the dataset, with increasing median throw in the South and decreasing in the North. The throw profile of Fault 3 (Fig. \ref{fig:07}b) shows two distinct levels of throw: In the Northern part of the fault median throw values are high at around $90$ to $105~m$, associated with comparatively lower uncertainty than in the South. Towards the South, median fault throw decreases down to about $40~m$ while the IQR values increase, being largest at the Southern edge of the dataset.

**10 (9):** How many students interpreted the fault further to the East?

**AUTHORS COMMENT**: Three students interpreted Fault 3 further to the East.

**CHANGE IN MANUSCRIPT**: -

**10 (11) to 11 (7):** This section suffers from a lack of statistical analysis, a framework to describe these results would increase the rigor of this section. The data shows some very important trends, probably the most important point of the manuscript, and with a more robust statistical analysis the reader would have more confidence in the results and following discussion.

**AUTHORS COMMENT**: We have increased the rigor in our statistical description of the results to increase clarity. Thank you for pointing this out!

**CHANGE IN MANUSCRIPT**: Several changes throughout the paragraph – please see Section 3.3 in the revised manuscript or the change-tracked manuscript.

**4 – Discussion**

**11 (10-18):** You open this paragraph with a statement that you show that u/c is correlated to seismic reflector strength, then backtrack on line 13 to discuss human factors. I would suggest that either this paragraph is split and both sections elaborated, or that the topic sentence incorporates both concepts. See Major comment 2 11 (26): How strong is the Top Ness horizon? Does this effect how well it is interpreted?

**AUTHORS COMMENT**: We do not see this as much as backtracking from the first statement, but rather we are addressing the complexity involved: Fault interpretation uncertainty correlates with seismic reflector strength, as our study results show, but it is dependent on many more variables (multiple correlation).

**CHANGE IN MANUSCRIPT**:

**13(31-32):** How did Fossen and Hesthammer (1998) get their pole? What was there scale of observation (i.e. did they have the data to extrapolate the sinusoidal shape)? The work on this should be included in this part of the discussion.

**AUTHORS COMMENT**: Fossen and Hesthammer (1998) calculated the pole using Bingham analysis of data with some significant spread. Their study was conducted on the basis of "reprocessed 3D seismic (ST8511)".

**CHANGE IN MANUSCRIPT**: We have added information on the origin of the data to the manuscript.

**12 (Fig 9):** How do you define interpretation density; units should be added if applicable? Visually this figure could do with a general text size increase, with many areas of text being too small. I would also suggest a change of colour for the boxes in part (1).

**AUTHORS COMMENT**: The term density refers to probability density of the interpretations. This just means that the integral of the histogram sums up to 1 and is a dimensionless number. This normalization makes the histograms comparable despite fluctuations in fault stick counts across the cube.

**CHANGE IN MANUSCRIPT**: We have increased text sizes and modified colours in Figure 9 to improve legibility.

**15 (5-7):** This is a potentially important point and raises a very important question 'what order did students interpret the cube?' If students are spending more time on a certain area, where data is of better quality then there are more factors to consider in why your results are different. Also does the style of interpretation change with time?

I advise either that the key findings section be reduced to a summary (e.g. set of bullet points) and separate section added to explore the reasons behind the uncertainty, probably split into 'technical' (e.g. image quality) and 'human' (e.g. different mental models) and that appropriate literature be added to this discussion.

**AUTHORS COMMENT**: Indeed! For such an analysis the entire interpretation process would need to be captured (e.g. screen capturing) and analysed, which was not an available option for this study. We have therefore no information about the temporal evolution of student's interpretation.

**CHANGE IN MANUSCRIPT**: -

**14 (27-29):** I think it would be unwise to suggest normal distributions, even in areas of good seismic data. I suspect in nearly all cases the distributions will be skewed. Most faults display an asymmetric damage zone, and such will also show an asymmetric signature in seismic, should the flat tail be towards the hanging wall?

**AUTHORS COMMENT**: The skewness of a Normal distribution can be an important factor for the detailed modeling of structures and should be considered wherever possible. But to our knowledge no detailed study has ever been carried out in trying to identify geological controls on skewness and kurtosis for probability distributions describing faults (or other geological structures). Stochastic geomodeling also needs to achieve a difficult trade-off between the level of detail and computational feasibility, as the correct sampling from the joint distribution constructed from complex parameter distributions (e.g. fat-tailed distributions) becomes increasingly challenging given today's computational constraints (e.g. Betancourt 2017, 2018). Coupling knowledge about asymmetric damage zones with interpretation experiments would be a very interesting future research topic though!

**CHANGE IN MANUSCRIPT**: -

**15 (3):** I found this an underwhelming end to a really neat data set. Although the implications for machine learning are indeed relevant, I feel the MS is crying out for a conclusion section which ties the findings together and includes the 'next stages' in tackling uncertainty in 3D seismic interpretation. The section itself also seems somewhat out of the remit of this work, and could

conceivably either be reduced or cut to make space for a discussion into the reasons behind the results as suggested previously.

**AUTHORS COMMENT**: We have added a conclusion section to the manuscript.

**CHANGE IN MANUSCRIPT**: Please see the revised manuscript for the added conclusions section.

Please find additional minor comments/suggested text edits on the attached MS (many of which are included in the specific comments).

**AUTHORS COMMENT**: We have considered and incorporated numerous minor comments and text edits from the attached commented MS.

**CHANGE IN MANUSCRIPT**: Please see the revised manuscript or change-tracked manuscript.

**Author Response to RC2 (Awad Bilal)**

**Major comments responses**

**[1]** While reviewing the manuscript, there have been some instances where the scientific approach was either oversimplified or need further clarifications. For example,

**[1a]** The authors attributed uncertainty in fault and horizon interpretations to mainly image quality. However, it was not clear to me what does seismic quality means in the manuscript (i.e. low amplitude and homogenous reflectors? low amplitude, chaotic reflectors? or noisy seismic. etc.). In addition, there was no information about the seismic data in the materials and methods section. I appreciate the work done in section 3.3 (seismic data quality) but I worry about the use of RMS attribute (reflector strength) to assess seismic data quality. Strong seismic reflections do not necessary means high seismic quality. Clear definition of what the authors consider as high or poor seismic quality is therefore crucial.

I suggest the authors start by describing the seismic stratigraphy of the sequences and the seismic characteristics of the analysed Top Ness horizon. Areas of poor seismic quality can then be highlighted together with the possible reasons behind low quality data (e.g. depositional environment, inherited low quality i.e. processing. . . etc). In order to give the reader more confidence, I suggest including a new figure (as Fig. 2 or 3), comprising two, N-S and E-W seismic sections, showing the interpreted horizons and seismic quality.

**AUTHORS RESPONSE**: Thank you for pointing out the lack of definition of the term seismic data quality. We have added a paragraph to the manuscript to explain our definition of the term. As our manuscript is mainly concerned with a first-look statistical description of the uncertainties encountered in 3-D seismic interpretation, we think it out of scope of this work to do a detailed study on the sources of seismic data quality and how it affects interpretation uncertainty. Nevertheless, this would be a logical next step for future research.

**CHANGE IN MANUSCRIPT**: Throughout this work we make use of the term seismic data quality not in the strictly geophysical sense of quality factors surrounding seismic data acquisition and processing, but rather in the sense of the interpretability of the seismic data. If the seismic data lacks clear, continuous reflectors in a region, but shows a noisy image difficult to interpret – no matter what the source of this may be – we describe it as an area of low seismic data quality. Similarly, if reflector strengths are high and continuous (for horizon interpretation) or clearly offset (for fault interpretation), we speak of high seismic data quality.

**[1b]** The authors observed that uncertainties in the horizon (Top Ness; Fig. 8) interpretation are significantly reduced surrounding wells with a general trend of increasing uncertainties from west to east. I suggest that this is further analysed and discussed. Important aspects to be considered are; (1) what is the seismic characteristics of the interpreted horizon? (2) what is the structural configuration (faulted, non-faulted) of the horizon; (3) how confident the students/authors are with both formation tops and well locations; and (4) is there any uncertainty in formation tops that can be compared/correlated to the uncertainties analysed in this paper. Additional factors that might also have affected interpretation include the level of assistance to students, time spent in the interpretation, the methodology and order of interpretation and how familiar the students were with using Petrel.

The fact that uncertainty increases away from well location is a general statement and, in many cases, it is invalid unless other factors are considered. Horizon interpretation uncertainty away from well location largely depends on the continuity of the horizon (also dependent on the depositional environment) and structural complexities. There are many cases where uncertainty is significantly low for hundreds of kilometres away from well location. Other cases show high uncertainty in near to the well or even in the Formation tops of the well.

**AUTHORS RESPONSE**: We have expanded our discussion regarding the horizon uncertainty surrounding well tops and faults in the results section to address some of the seismic characteristics of the horizon we believe relevant to the patterns (1). We believe the structural configuration is sufficiently characterized in the manuscript (2).

(3) We have no information about the inherent uncertainty in the formation tops and well locations.

(4) We have no information about the inherent uncertainty in the welltops. As all students received the same welltops, thus our analysis can be seen as "anchored" on the uncertainty in the dataset they received.

You raise a very good point regarding that the horizon interpretation uncertainty away from wells depends on the continuity of the horizon. We have incorporated this aspect into our manuscript.

**CHANGE IN MANUSCRIPT**: Figure 8a shows the average Top Ness horizon basemap for all interpretations combined. Overall the horizon interpretations are increasing in depth from SE towards the NW of the domain. Figure 8b shows the associated standard deviation of the average Ness horizon interpretation, with an overlay of mean fault intersections and well locations. We observe large horizon uncertainties in vicinity to both Faults 1 and 2 throughout the dataset. An increase of horizon uncertainty is poignant at the southern end of the domain where Faults 1 and 2 are beginning to merge again. In the North the horizon uncertainty surrounding Fault 3 decreases rapidly with distance from the fault, with two welltops and packages of high reflector strength (see Fig. 9A) constraining the uncertainty. As the seismic data quality decreases towards the south (see Fig. 9B), the uncertainty in the horizon interpretation increases in the Eastern part of the dataset. Interpretation uncertainties are significantly reduced surrounding well locations in the western part of the study domain, where seismic reflectors of the Top Ness horizon are overall stronger and more continuous (see Fig. 9A). This pattern does not hold true in the East of the dataset, where reflector continuity is overall low and noise in the seismic dataset is high.

**[1c]** I worry about the uncertainty of fault throw. For example, Figure 7b. shows that uncertainty is higher at smaller median throw (southern part of the fault), while it is expected to see high uncertainty at larger throw value. The figure also shows that same amount of median throw can have both high and low uncertainties (see central part of the fault Y 6784000). This can also be observed in Fig. 7a. Generally, uncertainty in throw mainly depends on the uncertainty of the correlation of the stratigraphic markers in the hanging wall and footwall blocks of the fault. For example, uncertainty in fault throw will be low where the stratigraphic correlation between fault blocks is good independently of throw value. In more detail, the uncertainty in cut-off angle in each sides of the fault depends on many aspects such as structural complexity, fault plane definition on the seismic, seismic quality near faults. The new publication of Ze and Alves titled 'Impacts of data sampling on the interpretation of normal fault propagation and segment linkage' might be of interest.

**AUTHORS RESPONSE**: We do not see problem of small median fault throws having larger uncertainties in this dataset. As we have described in the manuscript, the throw of Fault 3 becomes

more uncertain towards the South. We have detailed the decrease in data quality from North to South, making both fault placement (shown in Fig 3A) and horizon placement (shown in Fig 7b) more uncertain, which explains how the uncertainty of fault throw also increases in this area. The lack of clear stratigraphic markers on both sides of Fault 3 in the South, and the existence of good stratigraphic markers in the North emphasize this (as seen in Fig 8A and B).

**CHANGE IN MANUSCRIPT**: -

**[2]** I believe that the manuscript would be greatly improved and give more confidence to the reader if the 'Key findings' section of the discussion part is discussed in more depth. The fact that uncertainty in seismic interpretation increases in areas of poor seismic quality and linkage zones is relatively well known and expected. I appreciate the work done on quantifying these uncertainties, but I advise that the results of the paper are discussed in more detail. One important aspect that could be highlighted is the reasoning behind the variability in seismic interpretations, taking into consideration other factors that might affect seismic interpretation. Human bias in seismic interpretation and the expertise of the interpreters can also be addressed. It would also be interesting to see whether the result would change if the interpretation was carried out by experienced seismic interpreters using seismic filters to improve seismic quality together with various attributes (i.e. the effect of the level of experience). The authors can also analyse the effect of training (supervised top Cretaceous horizon interpretation) in reducing horizon interpretation uncertainty.

**AUTHORS RESPONSE**: We have overhauled the discussion / key findings part of our manuscript based on your and the other referees' comments.

Collecting and analysing extensive 3-D interpretations from experts would indeed form a great study but is definitely out of scope for this experiment. Also, we believe that the effect of training can not really be analysed from comparing the results presented in the manuscript with interpretations of the Top Cretaceous horizon, due to its stark contrast in structural simplicity and high reflector continuity compared to the Top Ness horizon throughout the dataset.

**CHANGE IN MANUSCRIPT**: Please see the revised manuscript or change-tracked manuscript for all the changes in the discussion.

**[3]** While reviewing the manuscript, there have been several instances where the description was not clear, or the authors use a great deal of qualitative language. I would prefer some quantifications of what few, major etc mean. These together with other minor corrections are highlighted in the attached original manuscript.

**AUTHORS RESPONSE**: Thank you for making us aware of this! We have removed the use of subjective language throughout the manuscript and have incorporated a large amount of your minor corrections.

**CHANGE IN MANUSCRIPT**: A multitude of changes throughout the manuscript. Please see the revised manuscript or the change-tracked manuscript.

**[4] Conclusions**: This paper describes a very important aspect of 3D seismic interpretation. However, a clear and concise summary with a clear link to the results and discussion would highlight the significance of the paper and results. Hence, I suggest the authors add a short summary or conclusions as bullet points with a clear take-home message.

**AUTHORS RESPONSE**: We have added a clear and concise conclusion to the manuscript.

**CHANGE IN MANUSCRIPT**: Please see the revised manuscript for the added conclusion section.

**[5] Figures**: Overall, the figures of the manuscript are clear and well-presented but please make sure of the resolution in the final printed version. Text size in some figures needs to be increased. Several figures have neither a scale bar nor north arrow. I also have suggested re-drafting and/or adding three new figures. Detailed comments on these new figures and on each figure are highlighted in the attached original manuscript.

**AUTHORS RESPONSE**: Thank you for pointing this out! We have increased text sizes and generally improved legibility of the figures throughout the manuscript. We have added scale bars and North arrows where applicable. We improved our Figures throughout the manuscript due to the reviewers input and merged the histogram and stereonet plots to improve the clarity for the reader. We decided against adding additional figures as we believe our improved Figures will adequately provide the reader with the relevant information and it would further bloat the manuscript. As our study mainly focuses on the structural uncertainties, we believe the example interpretation in combination with the average basemap and the seismic sections in the Figure 8 provide the reader with the context they need to follow the manuscript.

**CHANGE IN MANUSCRIPT**: Please the see the revised manuscript for changes in Figures.

**Detailed comments responses**

**Abstract**

Overall, it provides a clear summary but I believe it will be updated according to the reviewer's comments.

**AUTHORS RESPONSE**: We did not make changes to the summary, as the scientific content of the paper did not change.

**CHANGE IN MANUSCRIPT**: -

**Introduction**

Overall, the text is clear but I suggest the authors further clarify uncertainty in seismic interpretation and make it accessible to non-specialists. The authors could simply address that conceptual uncertainty mainly comes from human bias in seismic interpretation and depends on the expertise of the interpreter. Also, other types of uncertainties such as geophysical uncertainty, uncertainties related to formation tops and check-shot data could also be highlighted.

**AUTHORS RESPONSE:** We agree that a more specific description of what kind of uncertainty we are investigating (and how it related to "objective" uncertainty) would increase the clarity of the manuscript.

**CHANGE IN MANUSCRIPT:** Our work is thus concerned with quantifying the scope of uncertainties in seismic interpretation, which represents inevitably biased, human judgment under uncertainty (Tversky and Kahneman, 1974). This "subjective" uncertainty is in contrast to more "objective" uncertainty (Tannert et al., 2007; Bond, 2015) related to the geophysical acquisition of the data itself.

**[L12]** Please add the name of the 3D seismic survey

AUTHORS RESPONSE: We added the name and details of the 3-D seismic survey in the dataset section 2.1. Thanks for pointing that out!

CHANGE IN MANUSCRIPT: -

**[L14]** I would be more specific here. Seismic noise at depth? Near faults?

**AUTHORS RESPONSE:** We have expanded the specific description of seismic noise and referenced relevant Figure with seismic sections giving an example.

**CHANGE IN MANUSCRIPT:** The dataset depicts a comparatively simple geometry of planar domino-style normal faults, but the seismic dataset exhibits high amounts of noise, especially in its eastern half, and generally increasing with depth and in fault proximity (see Fig. 9).

**[L19]** The authors have shown a simple fault interpretation of Fossen and Hesthammer (1998) as a reference example. How about horizon interpretation? Do you have an interpreted horizon? Please find further comments on this below (i.e. Fig. 8).

**AUTHORS RESPONSE:** We use the fault interpretations from Fossen & Hesthammer (1998) as a conceptual reference interpretation – i.e. their fault network topology and the reported fault orientations – we do not have access to their unpublished interpretation. We clarified this in the manuscript.

**CHANGE IN MANUSCRIPT:** We use the interpretation of Fossen & Hesthammer (1998) as a reference expert example (in the sense of Macrae et al., 2016) to compare fault network topology and fault orientation uncertainty with the student interpretations.

**2.1 – Gullfaks geology and seismic data**

The paragraph lacks information about the 3D seismic data. Key information includes (1) the name of the 3D seismic survey; (2) acquisition year; (3) whether the data is originally in depth or time? and (4) key information about processing history as this will further clarify the seismic quality issue. I would also add the key tectonic phases evolution of the area (i.e. how many phases and ages).

**AUTHORS RESPONSE:** We agree that this information is of importance so assess sources of uncertainty within the seismic data itself. However, we have limited information on the dataset and its origin. We have added all relevant information that is available to us to the section.

We disagree on adding further detail on the tectonic evolution of the area into the paper and refer to the comprehensive description in published literature, as it would further bloat the manuscript.

**CHANGE IN MANUSCRIPT:** The 3-D seismic survey of the Gullfaks field, ST85R9211, was recorded in 1985 and reprocessed in 1992. It was recorded in time and converted to depth using TWT depth conversion and was migrated using a Prestack Kirchhoff migration

**L30-33** – Please clarify the research focus area both here and in Figure (1). I also suggest clarifying the scale of the interpretation area. Could you also highlight that the overall structure of the area is very complex particularly in the accommodation zone and collapsed footwall structures in the horst complex zone, while the western dominosystem (research focus) is comparatively simple.

**AUTHORS RESPONSE:** We have clarified the difference in complexity of the structural domains and improved clarity of the scale of the interpretation area in Figure 1.

**CHANGE IN MANUSCRIPT:** The field consists of three structurally distinct domains: a structurally simple domino system in the western part, and the structurally more complex accommodation zone and Horst complex towards the east (see Figure 1c). Our study focuses on the structurally simpler

western part of the domino system, where we investigate the uncertainty of the three faults F1-F3 and the fault blocks A-E depicted in Figure 1b.

**P3_Figure 1** - Please increase text size particularly in Fig. 1a and the Formation names in Fig. 1c. The location of the 3D seismic is not clear. The schematic fault map (b) is oversimplified. Please show fault displacement (fault polygons) and dip direction. Fault polygons would greatly help the reader to understand linkage zones as well as major and minor faults. I wonder how Fault 1 was identified in the northern part where it merges with Fault 2. Please see further comments highlighted on the attached original manuscript.

AUTHORS RESPONSE: We have improved the clarity of Figure 1. We have simplified the schematic fault map (Fig 1b) on purpose, as it was adapted from a heave-map of a different horizon from Fossen & Hesthammer (1998). We have added dip direction markers to the plot to improve clarity. We have shaded the schematic faults to make fault identification easier for the reader.

CHANGE IN MANUSCRIPT: Changes to Figure 1.

**2.2 – Interpretation dataset**

Many questions came to my mind with respect to seismic stratigraphy. What are the seismic characteristics of the Top Ness horizon and why it was selected for analysis? Please add this key information to give greater confidence to the reader with respect to the results.

AUTHORS RESPONSE: Our study focuses on the quantification of structural seismic interpretation uncertainties and we believe that a detailed discussion of seismic stratigraphy is out of scope for this work. The Top Ness horizon was interpreted by the students as part of their seismic interpretation course, thus the selection of the horizon was out of our hands. We agree though, that this would be very interesting topic for future research.

CHANGE IN MANUSCRIPT: -

**P3_L4** – Please show the location of the wells in Figure 1. How confident you are with well locations and Formation tops.

AUTHORS RESPONSE: We have no way to assess the uncertainty related to the well locations and formation tops. We agree that this information is vital to an ideal, uncertainty-integrated workflow, but our study also focuses on the "subjective" uncertainty produced by the interpretation. All students started out with the same set of well information.

CHANGE IN MANUSCRIPT: -

**P3_L8** – Why did students use manual interpretation? how often seeded tracking and manual interpretation were used?

**AUTHORS RESPONSE:** Due to the amount of noise throughout the dataset, manual interpretation would be necessary in many parts of the cube. We have no precise information about the frequency of individual interpretation tools. Our information is from the course instructions the students were given, and by interviewing a few students about their interpretation process. We have clarified this in the manuscript.

**CHANGE IN MANUSCRIPT:** Afterwards the students started to independently interpret the Base Cretaceous Unconformity (BCU) and Top Ness horizon (which is part of the Brent group) around well locations, followed by connecting the horizon interpretation in-between wells. The students were

instructed to mainly use guided auto-tracking, as well as occasional seeded tracking and manual interpretation where possible or necessary, depending on seismic data quality.

**P3_L9** – Do you mean interpolated auto-tracked surfaces? Did students actually use gridding? I would be more specific

**AUTHORS RESPONSE:** We removed the reference "gridded" as it can cause confusion in this context. It referred to the regularly gridded surfaces Petrel creates from horizon picks.

**CHANGE IN MANUSCRIPT:** The students then interpolated surfaces from the horizon interpretations using Petrel's Make Surface function.

**P4_L3** – Why the Top Ness horizon was selected for analysis. Why not BCU

**AUTHORS RESPONSE:** Due to work volume constraints in the course, students were instructed to only interpret the Top Ness horizon below the BCU. We chose to focus our analysis for this manuscript on the Top Ness horizon, due to its higher structural complexity compared to the BCU.

**CHANGE IN MANUSCRIPT:** -

**P4_Figure 2** - Please increase figure and text size. Also show both North arrow and a

scale bar.

**AUTHORS RESPONSE:** We have increased the text size and added North arrow and scale bar to the figure

**CHANGE IN MANUSCRIPT:** Figure 2 changes.

**2.3 – Data analysis**

It was not clear to me the key steps of the analysis. I would suggest adding a diagram showing steps of the workflow applied as this would greatly help the reader in particularly non-specialists. Please see further minor comments highlighted on the attached original manuscript.

**AUTHORS RESPONSE:** We decided against the addition of a workflow diagram, as we used common data analysis approaches to analyse our data and it would further bloat the manuscript. We provide our custom Python functionality in an open-source code repository for readers interested in the exact implementation. We hope our manuscript changes made particular parts of the data analysis clearer to the reader.

**CHANGE IN MANUSCRIPT:** Please see the revised or change-tracked manuscript for the changes made to the data analysis section.

**P5_L8** – I am not aware of RMSA. I am sure RMS only (without 'A') would be fine.

**AUTHORS RESPONSE:** We define RMSA as a shorthand for RMS Amplitude to use throughout the manuscript for brevity.

**CHANGE IN MANUSCRIPT: -**

**P5_6** - Figures 3 & 4 – I would suggest merging these figures or having both of them in the same page to make it easy for the reader to follow. Please make sure of the quality of the figure in the final printed version of the paper.

**AUTHORS RESPONSE:** Thank you for the suggestion! We have merged the two figures into one to improve clarity for the reader. Both figures are vector files, any quality issues might be due to pdf compression done by Copernicus or caused by the pdf-reader software used.

**CHANGE IN MANUSCRIPT:** Merged Figure 3 and 4 into Figure 3.

**P6_L2-3** – What do you mean by spread? Can you make this clear?

**AUTHORS RESPONSE:** Spread is used in its statistical meaning identical to scatter or dispersion.

**CHANGE IN MANUSCRIPT:** Changed spread to dispersion for clarity.

**P7_L13-14** – Why F3 was omitted? If it was interpreted in a similar fashion, why does uncertainty increases southwards as shown in Figure 3c?

**AUTHORS RESPONSE:** In all but a single student interpretation F3 does not interact with F1 or F2 and was thus not drawn in the fault network diagrams of the five most common fault network topologies, as it never changed. The uncertainty of F3's position and orientation increases towards the south, without impacting fault network topology.

**CHANGE IN MANUSCRIPT:** -

**P8_Figure 6** – Can you show students number in the Y-axis of Fig. 6a. Please add fault numbers (e.g. F1, F2) and dip direction in Fig. 6b. I was wondering how the authors differentiated between A and C and how fault 1 was defined in the northern part where it merges with fault 2 (Fig. 6b).

**AUTHORS RESPONSE:** We have added student numbers to the probability mass plot, as well as fault numbers. We think that dip directions would unnecessarily clutter the figure without adding critical information. We have added dip directions to the overview plot in Figure 1.

We differentiate A and C by the location of the fault stick placements in the dataset. North of the merging of Fault 1 and 2, the fault has been classified as belonging to the fault that terminates the other. E.g. if Fault 2 terminated at Fault 1, and Fault 1 continued northward, it was classified as Fault 1. We have enhanced the Figure schematics to improve clarity.

**CHANGE IN MANUSCRIPT:** Changes to Figure 6 (now Figure 5)

**2.3 – Fault throw and horizon uncertainty**

It was not clear to me whether the throw analysis was carried out along fault strike or a strike-parallel window.

What are the reasons behind selecting faults 1 and 3 (which was interpreted in a similar fashion and omitted in the previous section) for throw analysis?

**AUTHORS RESPONSE:** We have chosen to show the fault throw analysis for Faults 1 and 3 as the results of our analysis are reliable for them. The complex nature of Fault 2 and its subparts made it quite difficult for our algorithm to confidently analyse the fault throw. We are working on improving our algorithm for future work to apply it confidently to more complex faults. Another reason is, that Fault 2 and 3 are good examples of how fault throw uncertainty is changing across datasets.

**CHANGE IN MANUSCRIPT:** -

**P9_Figure 7a.** - How about uncertainty in the linkage zone? I think that the increase in uncertainty near survey edge is more likely related to the complex linkage zone to the south where faults 1, 2a and 2b merge as shown in Figure 1. It is well known that fault interaction can influence fault

propagation and hence, displacement profiles. Do you think that the displacement profile shown in Fig. 7a would change if fault 2 was added to fault 1. It will be very interesting to see both faults.

**AUTHORS RESPONSE:** We have modified Figure 7 to represent the data as boxplots along fault strike, which much better visualizes the changes in fault throw and its uncertainty, especially in the linkage zone of F1 and F2. Thank you for pointing out the linkage zone to the south – we've added this to the results.

**CHANGE IN MANUSCRIPT:** Changes in Figure 7 and throughout the manuscript.

**P9_Figure 7b**. – As indicated in the general comments above, this figure shows higher uncertainty associated with small median throw. Also, same amount of median throw show both high and low uncertainty. Please make of the analysis. Please see further comments highlighted in the attached original manuscript.

**AUTHORS RESPONSE:** We have redrafted Figure 7 as boxplots to better visualize the fault throw data. Fault throw in the Southern part of Fault 3 does show lower median with higher uncertainty (IQR), which is not unrealistic. We speculate this to be due to the problematic seismic data quality (in terms of interpretability), thus the variation in where students put the fault and their Top Ness horizon is increased (as seen also in Figure 8 surrounding Fault 3).

**CHANGE IN MANUSCRIPT:** Please see the change-tracked manuscript for the full amount of changes surrounding fault throw.

**P9_L4-9** - One key question came to my mind while reading the last paragraph of page 9 (description of Figure 8) is that where is the actual interpretation of the Top Ness Formation. I strongly suggest that the best interpreted horizon is added. so the reader can see the actual interpretation of both horizon and faults. It would be even better if it was interpreted by experienced seismic interpreter.

**AUTHORS RESPONSE:** We have to disagree with adding the best interpreted horizon, as we believe this is not very meaningful in this context. We have a large number of interpretations, and a priori no interpretation is necessarily better than any other as we lack knowledge of the real subsurface situation. This is why we present the mean Top Ness horizon in Figure 8a: to visualize a single interpretation in combination with the uncertainty in Figure 8b.

**CHANGE IN MANUSCRIPT: -**

P10_Figure 8 – Please redraft to include (a) the actual (best) interpreted horizon, (b) average horizon, and (c) the standard deviation horizon. Also, please add north arrow and scale bar in all parts of the figure. Please see further comments highlighted in the attached original manuscript.

**AUTHORS RESPONSE:** Please see the above response in regards why we believe that adding a single best horizon is not meaningful in the context of this manuscript.

**CHANGE IN MANUSCRIPT:** Added scale bar and North arrow to Figure 8.

**3.3 - Seismic data quality**

As indicated in the general comments above. I worry about the title of this section as well as the framework and results. The strength of seismic reflector does not necessarily mean good seismic quality. I also worry about the use of a time slice (Fig. 9.1 & 9.2) to assess seismic quality based on the strength of a single seismic reflector as the results can be significantly different at different levels or even the same level.

**AUTHORS RESPONSE:** We have chosen the time slice because we believe it serves as a valuable example of how the uncertainty in the interpretation of a single fault can vary in response to the seismic data quality (in terms of structural interpretability). As RMSA attribute is calculated across a vertical window, it includes information about reflectors above and below the depth slice shown.

**CHANGE IN MANUSCRIPT:** -

I believe that interpretation of fault 3 was aided in the northern part of the fault (Fig. 9.1 & inline section A) because of both the sharp reflection termination and the good correlation between seismic reflection packages on both sides of the fault. Generally, fault interpretation dose not actually depends on individual reflections, instead packages of reflections across the fault.

**AUTHORS RESPONSE:** This is the conclusion we are reaching in our manuscript: that sharp seismic reflector termination (as indicated by the through in RMSA response in Fig. 9a) strongly decreases fault interpretation uncertainty. We use Fault 3 to show how interpretation uncertainty increases when the noise of the dataset decreases and how we can possible use the RMSA response as a proxy for uncertainty.

**CHANGE IN MANUSCRIPT:** -

I would suggest the authors further test their results and see if the same approach can be applied to Fault 1 and 2 where the high amplitude to the south (Fig. 9) coincide with high uncertainty in both fault (Fig. 3) and horizon (Fig. 8) interpretation. Moreover, the authors could test whether the results are same if (1) the time slice was replaced by RMS attribute extraction on the actually interpreted horizon; or (2) a different time slice (e.g. 2100 or 2300) was used. Further minor comments on this and Figure 9 are highlighted in the attached original manuscript.

**AUTHORS RESPONSE:** We chose to present our analysis on Fault 3 as it serves as a meaningful example of the impact of noise / data quality on the interpretation uncertainty of a simple fault without many fault interactions. Data quality is generally poorer surrounding Faults 1 and 2 (as also seen in the increased spread in Figure 3) and the structural complexity is higher (merging and splitting of the faults). This is why we chose Fault 3 as a simple first example because Faults 1 and 2 are much more complex. We agree that there is much more opportunity for further research on how structural complexity interacts with fault interpretation uncertainty and data quality and we are aiming to do further work on this in the future.

(1) is a very good point! In more complex cases extracting the RMSA along a horizon probe would be ideal. But we lack an "actually interpreted" horizon (or rather we have too many valid ones). We could of course use the mean or median horizon, but this is not necessarily meaningful. As the pattern of increasing noise from North to South holds true for the dataset at all depths surrounding Fault 3 we believe it to be adequate to use a depth slice instead of a horizon probe at an arbitrary interpretation.

(2) We chose this specific depth slice as it encompasses easily interpretable results in the North with deceasing interpretability towards the South to show the trend visualized in Fig. 9a-d on how interpretation uncertainty correlates with seismic data quality.

**CHANGE IN MANUSCRIPT:** Several changes in section 3.3, please see the revised manuscript or difference-tracked manuscript.

**4 – Discussion**

Please see the attached original manuscript for further minor comments on the discussion part.

**AUTHORS RESPONSE**: We have incorporated many of your minor comments into the manuscript.

**CHANGE IN MANUSCRIPT**: Please see the revised and change-tracked manuscript for a detailed overview of all changes made.

[revised manuscript text omitted]